# ReasonEdit: Editing Vision–Language Models using Human Reasoning

**Jiaxing Qiu** [1]    **Kaihua Hou** [2]    **Roxana Daneshjou** [3]    **Ahmed Alaa** [2]    **Thomas Hartvigsen** [1]

## Abstract

Model editing aims to correct errors in large, pre-trained models without altering their unrelated behaviors. While some recent works have edited vision–language models (VLMs), no existing editors tackle reasoning-heavy tasks, which typically require humans and models to reason about images. We therefore propose ReasonEdit, the first VLM editor to let users explain their reasoning during editing, introducing a new, practical model editing setup. ReasonEdit continuously stores human reasoning in a codebook, and retrieves only relevant facts during inference using a novel topology-balanced multimodal embedding method inspired by network science. Across four VLMs on multiple rationale-based visual question answering datasets, ReasonEdit achieves state-of-the-art editing performance, ultimately showing that using human reasoning during editing greatly improves edit generalization. Our code and data are available at https://github.com/JiaxingQiu/reasonedit.

## 1. Introduction

Vision–language models (VLMs) have unlocked hard multimodal tasks like visual question answering (VQA) (Hartsock & Rasool, 2024; Shao et al., 2023; Khan & Fu, 2024), object recognition (Zhao et al., 2022; Du et al., 2022; Zang et al., 2025), and text-to-image generation (Koh et al., 2023; Zhao et al., 2024b;a). Frontier tasks now involve *reasoning*, where features in text and images are logically connected to the task (Khademi et al., 2023; Chen et al., 2024; Zhang et al., 2025b). For example, Figure 1 shows a real dermatology task, where visual features and domain knowledge combine to indicate the correct diagnosis. Despite steady

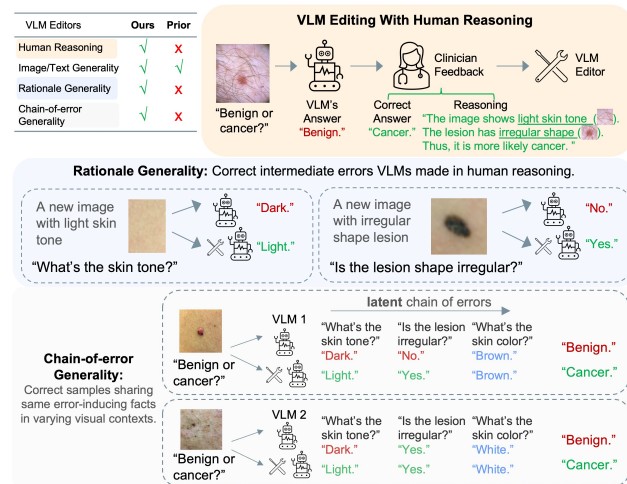

*Figure 1.* Reasoning-enhanced VLM editing lets users provide detailed feedback and reasoning, enabling broader generalization.

improvements, deployed VLMs will make errors as data distributions, labels, and user needs change over time, so we need ways to update models when errors occur.

Model editing has emerged to make efficient, targeted updates to large, pre-trained models, aiming at repairing a model's predictions with neither expensive retraining nor catastrophic decays to model performance (Wang et al., 2024; Gu et al., 2024; Zhang et al., 2025a). While most editing works study large language models (LLMs), particularly focusing on "knowledge," some editors have been developed for VLMs, with most simply applying LLM editing methods to the language model components within VLMs (Cheng et al., 2023; Huang et al., 2024; Zhang et al., 2025a; Zeng et al., 2025; Chen et al., 2025; Guo et al., 2025). However, these works neither address harder, more realistic VQA tasks that require detailed reasoning beyond renaming an object (Khademi et al., 2023; Cheng et al., 2024; Chen et al., 2024; Zhou et al., 2024), nor allow users to provide their own reasoning to produce more generalizable edits.

We propose editing VLMs using human reasoning: when a user queries a VLM and finds its response incorrect, they provide a chain of factual statements grounded in image or relevant knowledge that lead to the desired output. A reasoning-enhanced editor then uses these statements and the image-text query to produce a new VLM that (1) outputs the intended answer for the query, (2) generalizes the edit to

[1]University of Virginia, Charlottesville, VA, USA [2]University of California, Berkeley, Berkeley, CA, USA [3]Stanford University, Stanford, CA, USA. Correspondence to: Jiaxing Qiu <jq2uw@virginia.edu>, Thomas Hartvigsen <hartvigsen@virginia.edu>.

*Proceedings of the 43rd International Conference on Machine Learning*, Seoul, South Korea. PMLR 306, 2026. Copyright 2026 by the author(s).

sufficiently similar queries, and (3) preserves its predictions on unrelated inputs. Intuitively, using human reasoning adds highly relevant information during editing.

Leveraging human reasoning in VLM editing poses new challenges. First, as all model editors, our goal is to correct targeted errors at inference time (accuracy), generalize each edit to unseen samples (generality), and preserve unrelated behaviors (locality). Human reasoning enables new reasoning-enhanced generality by linking different samples through shared underlying reasoning, instead of only through semantic similarity in prompts. Second, human reasoning can include fine-grained visual details better represented by parts of an image than by the full, information-dense image (Fig. 1). This requires editors to align fine-grained visual details in the image with corresponding details in text, as acknowledged in prior work (Zhang et al., 2025a; Zeng et al., 2025). Third, reasoning statements are additional data to learn per edit. This suggests that weight-updating methods are more likely to suffer catastrophic degradation and further increase their already large computational costs (Mitchell et al., 2022; De Cao et al., 2021). Lastly, VLMs' architectures differ significantly from LLMs'. Yet model editors choose one or more layers to edit, a choice that is acknowledged to be simultaneously ad hoc and important (Cheng et al., 2023; Hartvigsen et al., 2023; Zeng et al., 2025). This choice is especially important for retrieval-based editors, which rely on semantically-similar edits being neighbors.

We propose ReasonEdit, the first reasoning-enhanced VLM editor. To align visual and textual details, ReasonEdit pairs each reasoning statement with image patches as visual evidence, which can be user-provided or VLM-approximated. During editing, ReasonEdit embeds image-text queries into a codebook as *keys* alongside human reasoning statements as *values*. During inference, the reasoning statements can then be selectively retrieved as appropriate context for future samples. By avoiding model weight updates and relying on retrieval, our solution avoids degrading the VLM's performance while maintaining low computational cost. To improve retrieval in multi-modal embedding space, we propose a novel topology-balanced multi-modal embedding method. Building on an observation that vision and language layers can each be biased towards their own modality, we propose treating multi-modal embeddings as nodes in a graph and measuring *modularity*, a widely-used measure of community-connectedness in network science. Using such embedding, queries and appropriate facts become neighbors, enabling better retrieval and highly generalizable edits.

In our experiments on four state-of-the-art VLMs using two human-rationale-included VQA datasets, ReasonEdit consistently achieves state-of-the-art performance on the standard model editing metrics: Our method successfully generalizes edits to similar images and questions without degrading model performance. It consistently demonstrates high performance and computing efficiency in the sequential editing, indicating its real-time usability with human feedback. Even further, we show that ReasonEdit induces new forms of generalization (Fig. 1), extending edits to samples that share underlying reasoning and to those where the same error-inducing facts appear in varying visual contexts.

In summary, our contributions are as follows:

1. We introduce the first reasoning-enhanced model editor, which we design for VLMs. This is the first model editing method to leverage human reasoning during editing. Our work suggests that collecting user feedback in practice may strongly improve model editing.

2. Our extensive experiments demonstrate that ReasonEdit is a state-of-the-art VLM editor that successfully leverages human reasoning to produce more-generalizable edits.

3. In proposing a principled way to select embeddings for image-text data, we also produce extensive insights into cross-modal relationships in VLM layer embeddings and how they relate to model editing, ultimately indicating this step is crucial to successful editing.

## 2. Related Work

**Model Editing.** Deployed large language models drift as knowledge and user needs change, motivating efficient, targeted editing methods that avoid expensive retraining. Early editors rely on auxiliary supervision for fine-tuning, which can be costly to obtain and risk overfitting (Sinitsin et al., 2020). More recent works improves editing precision by learning update rules via meta-learning (De Cao et al., 2021; Mitchell et al., 2021) or by locating and modifying localized factual components in the weights (Meng et al., 2022), with representative methods such as MEND (Mitchell et al., 2022) and ROME (Meng et al., 2022). Retrieval-based editors further limit adverse effects by storing edit information and reusing it selectively: GRACE (Hartvigsen et al., 2023) regularizes edits with a retrieval-based codebook, and IKE (Qi et al., 2024) retrieves relevant facts and prepends them as context, but assumes access to a representative edit distribution in advance. To broaden coverage across tasks, InstructEdit (Zhang et al., 2024b) uses instructions to guide a unified editor, but the instruction is limited to a set of specific tasks. However, none of these methods addresses editing on reasoning-heavy tasks beyond label correction, by allowing humans to provide detailed feedback.

**Model Editing for VLMs.** Recent benchmarks adapt LLM editors to VLMs and emphasize the multi-modal generality properties of VLM editors (Cheng et al., 2023; Huang et al., 2024; Zhang et al., 2025a). Recent works move toward

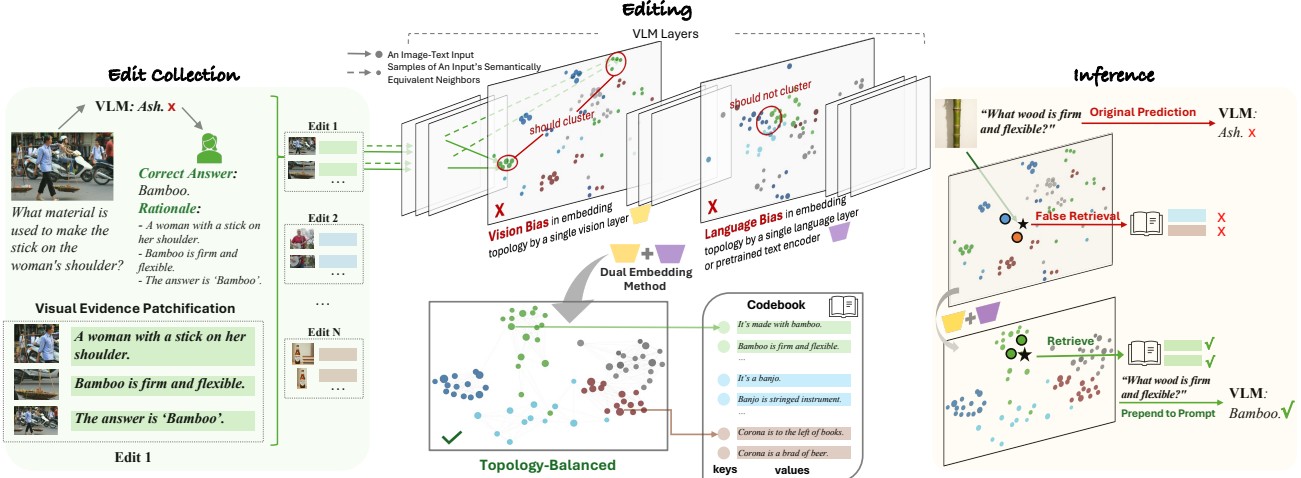

*Figure 2.* ReasonEdit allows users to provide detailed feedback when VLMs make errors in reasoning-heavy tasks. It converts each edit into image–text entries by pairing textual details from human reasoning with visual evidence image patches, and stores them in a codebook using a topology-aware multimodal embedding. At inference, it retrieves most relevant facts as a new query's context.

targeted VLM editing such as MSCKE (Zeng et al., 2025) that edits finer-grained visual entities. LiveEdit (Chen et al., 2025) avoids weight updates by routing to low-rank experts to retrieve labels, but requires pretraining on a complete edit set and per-edit generality and locality samples. BalancEdit (Guo et al., 2025) extends GRACE by explicitly balancing generality and locality in VLM editing. However, these methods do not support real-time editing in reasoning-heavy VQA tasks, where users correct errors by providing their own reasoning. Our work targets this gap by treating human reasoning as valuable insight into a VLM's generalizable failure modes, and retrieving correct facts for future samples where similar mistakes appear, thereby enabling broader generalization.

## 3. Reasoning-Enhanced VLM Editing

### 3.1. Problem Definition

Let $e = (i, t, y^+, r^+)$ denote an edit, where $i$ is an image, $t$ is a text question, $y^+$ is the corresponding text answer, and $r^+$ is a human-provided reasoning composed of $N_s$ step-by-step factual statements $r^+ = \{s_1, s_2, \ldots, s_{N_s}\}$ that lead to the correct answer. We aim to develop a model editor $g(\cdot)$ that takes in a pretrained VLM $f$ and an edit $e$, and outputs a new VLM $f_{\text{new}} = g(e, f)$. To align with real-world scenarios in which users provide feedback each time a new error occurs, an editor should support **sequential editing**, where it efficiently and continuously updates $f$ with each new edit. The edited VLM $f_{\text{new}}$ should respond correctly to edits without changing behavior on unrelated samples (locality). It should also generalize an edit to **unseen** samples that share semantic similarity in image or text (image-/text-generality), or rely on the same underlying reasoning facts (new forms of reasoning-enabled generality).

### 3.2. ReasonEdit

As illustrated in Fig. 2, we propose ReasonEdit, the first reasoning-enhanced VLM editor. ReasonEdit continuously engineers each reasoning-informed edit into detailed image-text entries, and stores them in a codebook to retrieve relevant facts during inference, without updating model weights.

**Visual Evidence Patchification.** As illustrated by the left example in Fig. 2, human reasoning may describe fine-grained visual details better captured by image parts than by the full image, requiring editors to align such visual details with their textual counterparts (Zhang et al., 2025a; Zeng et al., 2025). In ReasonEdit, users can provide cropped image patches as visual evidence for statements in their reasoning (e.g., Fig. 1). When no manual evidence is provided, we automatically identify image regions most relevant to a given sentence, using VLMs as an approximation. Given an image and a sentence $s_j \in r^+$, we generate candidate patches at multiple spatial scales and verify each patch by prompting the VLM with "Does the image show $s_j$?", retaining only patches whose predicted probability of "yes" exceeds that of "no". We score relevance using the log-likelihood of $s_j$ by prompting the VLM with the patch and the descriptive prompt: "Describe this image." Lastly, we select visual evidence by retaining the highest-likelihood patch overall and the highest-likelihood patch among the smallest candidates, balancing semantic relevance and spatial precision. Detailed algorithm of visual evidence patchification and a quantitative evaluation of its quality are provided in Appendix C. Fig. 10 contains an example.

**Codebook Construction.** During editing, ReasonEdit takes in a VLM $f$ and an edit $e = (i, t, y^+, r^+)$ at each time. As a retrieval-based editor, it sequentially constructs key–value entries $(K, V)$ per edit and adding them to a discrete codebook $\mathcal{C}$. Each key in the codebook is an embedding of

**Algorithm 1** Codebook Update from a New Edit

---

**Input:** $\mathcal{C} = \{(K_i, V_i)\}_{i=0}^{|\mathcal{C}|-1}$, codebook
**Input:** $f(\cdot)$, pretrained VLM
**Input:** $e = (i, t, y^+, r^+)$, edit
**Output:** $\mathcal{C}$, updated codebook
*Step 1: Add answer entry*
$K \leftarrow \mathcal{E}(i, t)$, $V \leftarrow$ sentence$(i, t, y^+)$
$\mathcal{C} \leftarrow$ merge keys $(\mathcal{C} \cup \{(K, V)\})$
*Step 2: Add rationale entries*
**for** each sentence $s_j \in r^+$ **do**
    Obtain associated visual evidence patches $\{i_j^p\}_{p=1}^P$
    **for** $p = 1$ **to** $P$ **do**
        $K \leftarrow \mathcal{E}(i_j^p, s_j)$, $V \leftarrow s_j$
        $\mathcal{C} \leftarrow$ merge keys$(\mathcal{C} \cup \{(K, V)\})$
    **end for**
**end for**
**return** $\mathcal{C}$

---

an image-text pair, denoted as $\mathcal{E}(i, t)$. For a given edit $e$, we generate two types of entries:

1. **An answer entry.** $(K = \mathcal{E}(i, t), V = \text{sentence}(t, y^+))$. The value sentence$(t, y^+)$ uses a fixed template: "The answer to question $t$ about the image is $y^+$." A similar query $(i_{\text{new}}, t_{\text{new}})$ will retrieve this correct answer and prepend it to the prompt.
2. **Reasoning entries.** For each $s_j \in r^+$, we first identify $P$ visual evidence image patches $\{i_j^p\}_{p=1}^P$ associated with fact $s_j$. Each patch–fact pair forms a reasoning entry $(\text{key} = \mathcal{E}(i_j^p, s_j), \text{value} = s_j)$, which stores the fine-grained visual and textual representation of a fact. A query related to this fact $\mathcal{E}(i_{\text{new}}, t_{\text{new}})$ will be close to $\mathcal{E}(i_j^p, s_j)$ and retrieve $s_j$ as supporting context.

To encourage edit generalization and minimize excessive codebook growth, ReasonEdit includes a key merging procedure. For each new key, we estimate a local neighborhood radius that captures semantic variability under small image and text perturbations. We then compare the new key with existing codebook entries and merge them when their neighborhoods substantially overlap. Specifically, to merge two keys, we require that the intersection area divided by the total area exceeds 90% for both radius-based regions, and that the embedding distance between their centers is smaller than 10% of both radii. Upon merging, the key is averaged and the value sentences are concatenated and unified; otherwise, the new key is added independently. Detailed algorithm of merging keys is provided in Appendix C. Over edits, the codebook $\mathcal{C} = \{(K_i, V_i)\}_{i=1}^{|\mathcal{C}|}$ is continuously updated via adding entries and merging, as described in Algorithm 1.

**Inference.** During inference, ReasonEdit retrieves relevant facts using a KNN procedure in the codebook's embedding space. Given a query $(i_{\text{new}}, t_{\text{new}})$, we compute its embedding $\mathcal{E}(i_{\text{new}}, t_{\text{new}})$ and measure its $\ell_2$ distance to all keys in the codebook. If the minimum query–key distance exceeds the $p$-th percentile of the pairwise distances among all keys, no retrieval is performed, as the query lies too far from the

existing edits. Otherwise, we retrieve $K$ nearest keys and collect all unique sentences from their values, and prepend them to the question as context in the final query.

### 3.3. Topology-Aware Multimodal Embedding

As with other retrieval-based editors, ReasonEdit's performance largely depends on the precision of retrieval in the embedding space. Previous editors embed image–text data using a single layer, a choice known to be important but made post hoc by ablating editing performance (Cheng et al., 2023; Hartvigsen et al., 2023; Zeng et al., 2025; Guo et al., 2025). We propose thinking of relationships between image-text embeddings as a graph, and introduce a principled topology-aware criterion to evaluate and select across embedding methods. As shown in Fig. 2, a multimodal embedding biased toward a single modality—clustering embeddings predominantly by either image or text similarity—leads to poor retrieval. Therefore, we use a novel topology-balanced dual embedding, denoted $\mathcal{E}_{\text{dual}}$, to serve as keys ($\mathcal{E}(i, t)$) in ReasonEdit's codebook.

**Embedding Evaluation by Graph Topology.** We propose a principled multimodal embedding evaluation strategy using five network-science-based metrics: vision modularity, language modularity, bimodal modularity, vision bias, and language bias. Details are provided in Appendix A; here we present the essential formulations.

In network science, ***Newman's modularity*** $Q$ measures how well a network aligns with a given partition (grouping of nodes) (Newman, 2004; 2006). Given an image $i$ and text $t$, let $z = \mathcal{E}(i, t) \in \mathbb{R}^d$ denote their embedding. Each embedding space induces a weighted similarity network $G = (V, A)$, where each node $z_u \in V$ corresponds to an image–text pair $(i_u, t_u)$ and each edge weight $A_{uv}$ measures similarity between two embeddings. We define $A_{uv} = \frac{d_{\max} - \|z_u - z_v\|_2}{d_{\max} - d_{\min}}$, $A_{uu} = 0$ where $d_{\min}$ and $d_{\max}$ are the minimum and maximum pairwise distances. This normalization ensures $A_{uv} \in [0, 1]$ and scale invariance of the embedding space, enabling direct comparison across embedding methods (see Appendix A.4 for proofs). Since it's impossible to construct a network over all possible image–text pairs, we approximate via Monte Carlo sampling. For each random batch of $n$ image–text pairs, we construct a sample network $G_n = (V_n, A_n)$ and compute modularity with respect to an expected partition $g_n$. Averaging over $B$ independent sample networks yields a *sample modularity*: $\widehat{Q} = \frac{1}{B} \sum_{b=1}^B Q\left(A_n^{(b)}; g_n^{(b)}\right)$. Higher $\widehat{Q}$ indicates stronger alignment between the embedding network and a partition.

We define the desired topology of such an embedding similarity network via ***expected partition***. It specifies the grouping of embeddings that correctly reflect the semantic structure of multi-modal data. Thereby, we can compute vision,

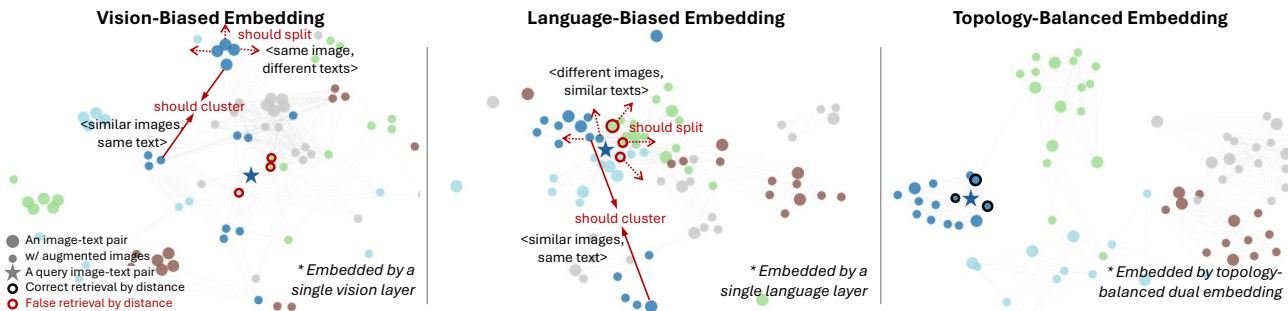

*Figure 3.* Visualization of embedding networks. Each color represents one image (and its augmentations) paired with varying texts. Vision-biased embeddings by a single vision layer cluster irrelevant texts due to image similarity. Language-biased embeddings produced by a language layer cluster different images due to text similarity. Such unimodal-biases lead to false distance-based retrievals. In contrast, The topology-balanced embedding aligns with joint image–text similarity. Demonstration with *InstructBLIP-Vicuna-7B*.

language and bimodal sample modularities as follows:

- Given a batch of $n$ image-text pairs, we form the embedding similarity network $A_{\text{vis}}$ over all cross-product image–text pairs $\langle i_a, t_b \rangle$ with $a, b \in \{1, \ldots, n\}$. The expected vision partition groups nodes by image identity: $g_{\text{vis}}(\langle i_a, t_b \rangle) = a$. The **vision modularity** is $\widehat{Q}_{\text{vis}} = \frac{1}{B} \sum_{b=1}^{B} Q(A_{\text{vis}}; g_{\text{vis}})$. A high $\widehat{Q}_{\text{vis}}$ reflects alignment with image-only similarity.

- We define **language modularity** in the same fashion as vision modularity, except that the expected partition groups nodes by text identity: $g_{\text{lang}}(\langle i_a, t_b \rangle) = b$. A high $\widehat{Q}_{\text{lang}}$ reflects alignment with text-only similarity.

- We define **bimodal modularity** analogously, where each image–text pair and its augmentations form a cluster. A higher $\widehat{Q}_{\text{bi}}$ indicates nodes highly similar under both modalities are assigned to the same cluster, while nodes dissimilar in either modality are separated.

To quantify misalignment between an embedding and the desired topology, we introduce two topological biases that can harm distance-based retrieval by over-relying on a single modality. We diagnose $\mathcal{E}(\cdot)$ as **vision-biased** if it places (same image, different text) pairs closer than (similar image, same text) pairs to a given image–text pair. For retrieval-based editors, large positive vision bias causes over-sensitivity to image perturbations, yielding poor image generality as augmented images are pushed away from their expected clusters, and worse locality as irrelevant texts are retrieved for the same image. **Language bias** is defined analogously. Large positive language bias indicates over-sensitivity to text perturbations, leading to poor text generality and degraded locality due to retrieval from irrelevant images. Detailed calculations are in Appendix A. Fig. 3 illustrates vision-biased embeddings generated by a single vision-block layer and language-biased embeddings generated by a single language layer of a VLM.

**Topology-Balanced Dual Embedding.** To reduce unimodal biases in single-layer embeddings, we propose a topology-balanced dual embedding method, $\mathcal{E}_{\text{dual}}$, which concate-

nates a single vision-layer embedding with a pretrained sentence-transformer text embedding for an image–text pair $(i, t)$. We select the vision layer with mean pooling embedding by maximizing bimodal sample modularity across all vision layers $\mathcal{L}_{\text{vision}}$:

$$\mathcal{E}_{\text{dual}}(i, t) = \left[ \mathcal{E}_{\text{vision}}^{(l^*)}(i, t); \; w \cdot \mathcal{E}_{\text{sbert}}(t) \right], \; l^* = \arg\max_{l \in \mathcal{L}_{\text{vision}}} \widehat{Q}_{\text{bi}}^{(l)}.$$

The weight $w$ controls the relative scale of the text-only dimensions: larger $w$ enforces stronger language topology alignment (and language bias), while smaller $w$ enforces stronger vision topology alignment (and vision bias). As varying $w$ induces a vision–language modularity trade-off, we select $w$ by maximizing their harmonic mean: $w^* = \arg\max_w \frac{2 \widehat{Q}_{\text{vis}}(w) \widehat{Q}_{\text{lang}}(w)}{\widehat{Q}_{\text{vis}}(w) + \widehat{Q}_{\text{lang}}(w)}$. This dual embedding mitigates the vision bias of a single vision-layer embedding by augmenting it with a pretrained text embedding space, as sentence-transformer encoders produce embeddings that align well with semantic similarity in text (Reimers & Gurevych, 2019).

## 4. Experiments

### 4.1. Evaluation Metrics

Based on the problem definition, in addition to conventional properties, we propose two new reasoning-enhanced generalities that $f_{\text{new}}$ should achieve: *rationale generality* and *chain-of-error generality*.

**(1) Reliability (Acc).** The updated model should output the correct answer for the set of edits $D_{\text{edit}}$. Reliability $= \mathbb{E}_{(i,t,y^+,r^+) \in D_{\text{edit}}} \mathbf{1}\{f_{\text{new}}(i, t) = y^+\}$.

**(2) Locality (Loc).** The updated model should retain the original predictions on unrelated samples that were previously answered correctly. Let $D_u$ denote the set of unrelated samples associated with edit $(i, t)$. Locality $= \mathbb{E}_{(i',t') \in D_u} \mathbf{1}\{f_{\text{new}}(i', t') = f(i', t')\}$. Note that the overall post-edit accuracy is the combination of Acc and Loc.

**(3) Text Generality (T-Gen).** The updated model should output the correct answer when given new rephrased

versions of the original question. Let $R(t)$ denote the set of rephrased versions of question $t$. T-Gen $= \mathbb{E}_{(i,t,y^+,r^+)\in D_{\text{edit}}}\mathbf{1}\{f_{\text{new}}(i, R(t)) = y^+\}$.

**(4) Image Generality (I-Gen).** The updated model should output the correct answer when the original image is replaced by a semantically similar new image. Let $R(i)$ denote similar images to $i$. I-Gen $= \mathbb{E}_{(i,t,y^+,r^+)\in D_{\text{edit}}}\mathbf{1}\{f_{\text{new}}(R(i),t) = y^+\}$.

**(5) Rationale Generality (R-Gen).** The updated model should correctly answer questions about any intermediate facts provided in the human reasoning $r^+ = \{s_1, s_2, \ldots, s_{N_s}\}$. Let $S \subseteq r^+$ be any non-empty subset of reasoning facts, and let $(i_S, t_S)$ denote a new image–question pair that relies on $S$, with correct answer $y_S^+$. We define *rationale generality* as R-Gen $= \mathbb{E}_{(i,t,y^+,r^+)\in D_{\text{edit}}, S\subseteq r^+}\mathbf{1}\{f_{\text{new}}(i_S,t_S) = y_S^+\}$. Fig. 8 shows examples of the R-Gen evaluation dataset.

**(6) Chain-of-Error Generality (CoE-Gen).** We ask the VLM to recognize each fact in the human reasoning; those it fails to recognize are defined as error-inducing facts. The updated model should answer the original question correctly when the same error-inducing facts appear in a different visual context. Let $i_{\text{coe}}$ denote new images in which the error-inducing facts are visually present. CoE-Gen $= \mathbb{E}_{(i,t,y^+,r^+)\in D_{\text{edit}}}\mathbf{1}\{f_{\text{new}}(i_{\text{coe}},t) = y^+\}$. Fig. 9 shows examples of the CoE-Gen evaluation dataset.

Note that all evaluation datasets are unseen new samples. Details of the evaluation datasets construction are provided in Appendix B.

### 4.2. Experimental Setup

**Datasets.** We use two public rationale-VQA datasets, where each image–question–answer tuple is paired with human-written reasoning that explains how image features relate to each other and the answer. (1) A-OKVQA (Schwenk et al., 2022) contains 18,195 questions for 17,656 COCO-2017 images (Chen et al., 2015). (2) FVQA (Wang et al., 2017) contains 5,826 questions over 2,190 images from COCO-2014 (Chen et al., 2015) and ImageNet-2012 (Russakovsky et al., 2015). We follow these prior works to prompt VLMs with the original image–question pair and select the final answer from the multiple-choice option with the highest probability, yielding stable answer label predictions. Thus, in our experiments, the edit set $D_{\text{edit}}$ contains real errors made by VLMs, and the unrelated set $\mathcal{D}_{\text{u}}$ contains correctly-answered samples. VLM generation is deterministic to ensure reproducibility. See details in Appendix D.1.

**Edited VLMs.** We edit four state-of-the-art reasoning VLMs with diverse architectures and varying sizes: QWEN3-VL-4B-INSTRUCT , QWEN3-VL-8B-INSTRUCT (Yang et al., 2025), INSTRUCTBLIP-7B (Dai et al., 2023), and LLAVA-

1.5-7B (Liu et al., 2023). Details about these VLMs are provided in Appendix D.2.

**Baseline Model Editors.** We compare ReasonEdit with five state-of-the-art editors adapted or proposed for VLM editing. For weight-updating editors: (1) FT. We finetune selected layers of the VLM for each edit (Guo et al., 2025). (2) MEND (Mitchell et al., 2021). A meta-learning editor that predicts weight updates for future edits. (3) BalancEdit (Guo et al., 2025). A VLM editor that learns a dynamic influence radius and applies localized weight updates. For retrieval-based editors: (4) IKE (Qi et al., 2024; Guo et al., 2025). A retrieval-based editor that prepends relevant facts to the prompt. (5) GRACE (Hartvigsen et al., 2023). A lifelong editor that stores edits in a codebook and applies them near past errors. Beyond their original use, we augment all editors to be reasoning-enabled (denoted editor-COT) so that they receive the same inputs as ReasonEdit. For weight-updating editors, we train the model to generate the correct answer followed by human reasoning. For retrieval-based GRACE and IKE, we populate the codebook with entries for all factual sentences in the human reasoning. Details on editors and their selection are provided in Appendix D.3.

**Implementation Details.** We concatenate a pretrained sentence encoder (paraphrase-mpnet-base-v2 (Reimers & Gurevych, 2019)) with the vision layer (chosen by sample modularity) in ReasonEdit's dual embedding. Sample modularity is estimated via Monte Carlo over $B = 10$ independent sample networks, which is sufficient for variance to converge with low computational cost. The $n$ image–text pairs used to construct each network are sampled from the combined dataset of COCO and ImageNet. We find the topology-aware embedding selection strategy is **data-agnostic**, when selecting $l$ and $w$ using three different datasets: COCO (Chen et al., 2015), ImageNet (Russakovsky et al., 2015), and Flickr30k (Plummer et al., 2015). Details are provided in Appendix A.3. We try $n = 5, 10, 20$ (Supplemental Fig. 13) and pick $n = 10$ for both low Monte Carlo variance and low computational cost. At inference, we retrieve $K = 5$ nearest keys and use a no-retrieval threshold of $p = 50$-th percentile for all VLMs. Ablations are presented in Sec. 4.6. Prior editors are configured as closely as possible to their original implementations while ensuring fair comparisons; details are provided in Appendix D.4.

### 4.3. Comparisons to Prior Editors

We first consider static editing, where a set of edits is made available to each editor. Then the edited model is evaluated on a set of unseen edits. As shown in Table 1, in editing errors made by each VLM on two datasets, ReasonEdit consistently outperforms prior editors and their reasoning variants, across editing metrics. Most importantly, ReasonEdit significantly improves rationale and CoE generality, while prior editors marginally exceed the baseline. This

| Editor | LLaVA-1.5-7B (811 Errors) | | | | | | InstructBLIP-7B (764 Errors) | | | | | | Qwen3-VL-8B (857 Errors) | | | | | | Qwen3-VL-4B (700 Errors) | | | | | |
|---|---|---|---|---|---|---|---|---|---|---|---|---|---|---|---|---|---|---|---|---|---|---|---|---|
| | Acc | T-Gen | I-Gen | CoE-Gen | R-Gen | Loc | Acc | T-Gen | I-Gen | CoE-Gen | R-Gen | Loc | Acc | T-Gen | I-Gen | CoE-Gen | R-Gen | Loc | Acc | T-Gen | I-Gen | CoE-Gen | R-Gen | Loc |
| Unedited | 0.00 | 0.17 | 0.52 | 0.45 | 0.76 | 1.00 | 0.00 | 0.25 | 0.52 | 0.54 | 0.80 | 1.00 | 0.00 | 0.16 | 0.47 | 0.40 | 0.74 | 1.00 | 0.00 | 0.18 | 0.48 | 0.43 | 0.73 | 1.00 |
| FT | 0.83 | 0.76 | 0.69 | 0.70 | 0.39 | 0.53 | 0.78 | 0.64 | 0.68 | 0.56 | 0.40 | 0.57 | 0.64 | 0.60 | 0.51 | 0.58 | 0.27 | 0.36 | 0.69 | 0.64 | 0.53 | 0.58 | 0.27 | 0.38 |
| MEND | 0.52 | 0.50 | 0.45 | 0.49 | 0.26 | 0.40 | 0.28 | 0.25 | 0.27 | 0.26 | 0.22 | 0.34 | 0.33 | 0.33 | 0.29 | 0.29 | 0.19 | 0.27 | 0.44 | 0.42 | 0.37 | 0.41 | 0.20 | 0.29 |
| IKE | 0.87 | 0.80 | 0.86 | 0.87 | 0.79 | **1.00** | 0.93 | 0.87 | **0.95** | 0.86 | 0.75 | **1.00** | 0.92 | 0.80 | **0.97** | 0.70 | 0.78 | **1.00** | 0.90 | 0.76 | 0.86 | 0.70 | 0.75 | **1.00** |
| GRACE | 0.98 | 0.69 | 0.92 | 0.54 | 0.76 | 0.99 | **0.98** | 0.59 | 0.91 | 0.56 | 0.80 | 0.99 | **0.98** | 0.79 | 0.79 | 0.60 | 0.75 | 0.86 | **0.98** | 0.77 | 0.82 | 0.60 | 0.73 | 0.93 |
| BalancEdit | 0.83 | 0.79 | 0.53 | 0.47 | 0.47 | 0.49 | 0.85 | 0.62 | 0.64 | 0.47 | 0.55 | 0.60 | 0.66 | 0.64 | 0.40 | 0.41 | 0.32 | 0.31 | 0.73 | 0.71 | 0.38 | 0.45 | 0.32 | 0.39 |
| FT-COT | 0.85 | 0.77 | 0.75 | 0.71 | 0.61 | 0.66 | 0.84 | 0.68 | 0.79 | 0.61 | 0.67 | 0.71 | 0.68 | 0.65 | 0.57 | 0.62 | 0.49 | 0.49 | 0.78 | 0.71 | 0.62 | 0.66 | 0.50 | 0.48 |
| MEND-COT | 0.45 | 0.41 | 0.53 | 0.45 | 0.49 | 0.59 | 0.50 | 0.44 | 0.58 | 0.45 | 0.51 | 0.66 | 0.38 | 0.38 | 0.37 | 0.34 | 0.34 | 0.38 | 0.36 | 0.33 | 0.29 | 0.33 | 0.33 | 0.32 |
| IKE-COT | 0.88 | 0.79 | 0.86 | 0.85 | 0.81 | **1.00** | 0.93 | 0.85 | 0.94 | 0.84 | 0.80 | **1.00** | 0.92 | 0.78 | 0.90 | 0.77 | 0.79 | **1.00** | 0.91 | 0.73 | 0.87 | 0.69 | 0.78 | **1.00** |
| GRACE-COT | 0.98 | 0.69 | 0.92 | 0.54 | 0.76 | 0.99 | **0.98** | 0.57 | 0.91 | 0.57 | 0.80 | 0.98 | **0.98** | 0.79 | 0.79 | 0.60 | 0.75 | 0.86 | **0.98** | 0.77 | 0.82 | 0.60 | 0.73 | 0.86 |
| BalancEdit-COT | 0.86 | 0.80 | 0.60 | 0.57 | 0.60 | 0.71 | 0.55 | 0.49 | 0.61 | 0.61 | 0.73 | 0.88 | 0.40 | 0.40 | 0.39 | 0.41 | 0.60 | 0.42 | 0.46 | 0.45 | 0.43 | 0.41 | 0.57 | 0.52 |
| **ReasonEdit** | **1.00** | **0.97** | **0.95** | **0.97** | **0.87** | **1.00** | **0.96** | **0.94** | 0.92 | **0.92** | **0.88** | **1.00** | **0.96** | **0.85** | 0.92 | **0.95** | **0.81** | **1.00** | **0.98** | **0.85** | **0.91** | **0.95** | **0.83** | **1.00** |

(a) Dataset 1: FVQA

| Editor | LLaVA-1.5-7B (7197 Errors) | | | | | | InstructBLIP-7B (5954 Errors) | | | | | | Qwen3-VL-8B (7139 Errors) | | | | | | Qwen3-VL-4B (5188 Errors) | | | | | |
|---|---|---|---|---|---|---|---|---|---|---|---|---|---|---|---|---|---|---|---|---|---|---|---|---|
| | Acc | T-Gen | I-Gen | CoE-Gen | R-Gen | Loc | Acc | T-Gen | I-Gen | CoE-Gen | R-Gen | Loc | Acc | T-Gen | I-Gen | CoE-Gen | R-Gen | Loc | Acc | T-Gen | I-Gen | CoE-Gen | R-Gen | Loc |
| Unedited | 0.00 | 0.12 | 0.10 | 0.37 | 0.61 | 1.00 | 0.00 | 0.14 | 0.11 | 0.48 | 0.63 | 1.00 | 0.00 | 0.12 | 0.11 | 0.33 | 0.58 | 1.00 | 0.00 | 0.14 | 0.15 | 0.38 | 0.58 | 1.00 |
| FT | 0.59 | 0.55 | 0.53 | 0.53 | 0.34 | 0.40 | 0.54 | 0.46 | 0.49 | 0.47 | 0.33 | 0.52 | 0.48 | 0.45 | 0.44 | 0.45 | 0.24 | 0.28 | 0.37 | 0.35 | 0.33 | 0.33 | 0.22 | 0.30 |
| MEND | 0.62 | 0.58 | 0.59 | 0.59 | 0.28 | 0.35 | 0.53 | 0.47 | 0.48 | 0.51 | 0.27 | 0.33 | 0.28 | 0.28 | 0.29 | 0.27 | 0.19 | 0.22 | 0.25 | 0.25 | 0.23 | 0.25 | 0.20 | 0.28 |
| IKE | 0.97 | 0.89 | 0.95 | 0.82 | 0.62 | **1.00** | 0.98 | 0.92 | **0.98** | 0.85 | 0.67 | **1.00** | 0.99 | 0.86 | **0.97** | 0.75 | 0.58 | **1.00** | 0.95 | 0.87 | 0.95 | 0.79 | 0.57 | **1.00** |
| GRACE | 0.97 | 0.84 | 0.89 | 0.64 | 0.58 | 0.99 | 0.98 | 0.73 | 0.87 | 0.62 | 0.63 | 0.99 | 0.98 | 0.86 | 0.83 | 0.69 | 0.60 | 0.87 | 0.97 | 0.85 | 0.84 | 0.71 | 0.59 | 0.87 |
| BalancEdit | 0.73 | 0.69 | 0.28 | 0.32 | 0.49 | 0.60 | 0.59 | 0.46 | 0.39 | 0.35 | 0.44 | 0.56 | 0.56 | 0.54 | 0.37 | 0.47 | 0.42 | 0.45 | 0.41 | 0.39 | 0.22 | 0.24 | 0.31 | 0.36 |
| FT-COT | 0.71 | 0.66 | 0.64 | 0.63 | 0.52 | 0.50 | 0.69 | 0.55 | 0.58 | 0.59 | 0.52 | 0.53 | 0.61 | 0.58 | 0.55 | 0.58 | 0.44 | 0.35 | 0.61 | 0.58 | 0.52 | 0.56 | 0.41 | 0.35 |
| MEND-COT | 0.52 | 0.49 | 0.46 | 0.47 | 0.45 | 0.46 | 0.46 | 0.42 | 0.41 | 0.43 | 0.41 | 0.43 | 0.26 | 0.26 | 0.25 | 0.26 | 0.25 | 0.29 | 0.23 | 0.24 | 0.22 | 0.26 | 0.26 | 0.30 |
| IKE-COT | 0.96 | 0.88 | 0.95 | 0.85 | 0.76 | **1.00** | 0.97 | 0.91 | **0.98** | 0.85 | 0.77 | **1.00** | 0.99 | 0.83 | **0.97** | 0.74 | 0.70 | **1.00** | 0.96 | 0.87 | **0.96** | 0.78 | 0.66 | **1.00** |
| GRACE-COT | **0.99** | 0.85 | 0.91 | 0.62 | 0.61 | 0.98 | 0.97 | 0.77 | 0.88 | 0.58 | 0.64 | 0.98 | **0.99** | 0.88 | 0.89 | 0.76 | 0.61 | 0.90 | **1.00** | **0.91** | 0.93 | 0.80 | 0.60 | 0.95 |
| BalancEdit-COT | 0.73 | 0.68 | 0.25 | 0.29 | 0.54 | 0.73 | 0.31 | 0.23 | 0.28 | 0.48 | 0.62 | 0.92 | 0.23 | 0.25 | 0.24 | 0.35 | 0.55 | 0.63 | 0.27 | 0.26 | 0.23 | 0.25 | 0.52 | 0.71 |
| **ReasonEdit** | **0.99** | **0.97** | **0.98** | **0.98** | **0.84** | **1.00** | **0.99** | **0.97** | **0.98** | **0.99** | **0.86** | **1.00** | 0.98 | **0.92** | **0.97** | **0.95** | **0.79** | **1.00** | 0.97 | 0.90 | **0.96** | **0.96** | **0.77** | **1.00** |

(b) Dataset 2: A-OKVQA

*Table 1.* Editing results on FVQA and A-OKVQA datasets. Best results are in bold.

shows that ReasonEdit enables new reasoning-induced generalization to diverse samples that rely on shared human reasoning. ReasonEdit achieves strong locality due to its retrieval-based design, similar to IKE and GRACE. It attains higher image and text generality than prior editors by using topology-balanced dual embeddings as codebook keys instead of single-layer embeddings (see ablations Sec.4.6). COT-variants of prior editors also overall yield small gains in R-Gen and CoE-Gen, but their absolute performance remains low and close to the unedited model. This shows that merely adding reasoning supervision is insufficient for reasoning-induced generalization; ReasonEdit succeeds by retrieving appropriate human reasoning.

### 4.4. Sequential Editing and Computing Efficiency

We next consider sequential editing, where each editor performs one edit at a time to repeatedly edit the same VLM. The edited model is then evaluated every 200 edits on a random batch of 50 accumulated edits. We use A-OKVQA, which contains over 5,000 edits for all VLMs. As shown on the left side of Figure 4, ReasonEdit achieves stable and consistently-higher editing performances than other editors. Retrieval-based GRACE and IKE show second-best reliability, image generality, text generality, and locality over edits. The declining accuracy, image generality, and text generality of weight-updating editors (FT, MEND, and BalancEdit) indicate catastrophic model degradation over a larger number of edits. The drop in BalancEdit's locality suggests that its radius regularization, derived from the generality–locality trade-off, is effective under a small number of edits. On the

right side of Figure 4, we also show the time taken per editor on the same hardware. Overall, ReasonEdit takes the same amount of time as the non-COT editors, but is significantly faster than their COT extensions (see Supplemental Fig. 12).

### 4.5. Robustness under Noisy Reasoning

ReasonEdit assumes users provide accurate and structured reasoning, but human reasoning can be noisy, incomplete, or incorrect in real scenarios. We therefore evaluate the robustness of ReasonEdit under noisy reasoning. During editing, for each edit, we either (1) inject or (2) replace 10%, 30%, and 50% of its reasoning statements with noise (irrelevant random sentences). As shown in Table 2, ReasonEdit is highly robust to injected noise, with performance remaining close to the noise-free setting. When real reasoning facts are replaced with noise, R-Gen and CoE-Gen degrade as expected, since the codebook no longer contains all relevant facts. This also highlights the crucial role of reasoning facts in enabling reasoning-level generalization in model editing.

### 4.6. Ablation Studies

We extensively ablate ReasonEdit to examine its sources of performance, using the combination of both datasets.

**(1) Isolating answer and reasoning effects on editing performance.** We first ablate the effects of answers and reasoning on editing performance, by keeping only reasoning entries or only answer entries in the codebook. As shown in Table 3, using answer entries, ReasonEdit's performance is close to the upper bound where the ground-truth (GT)

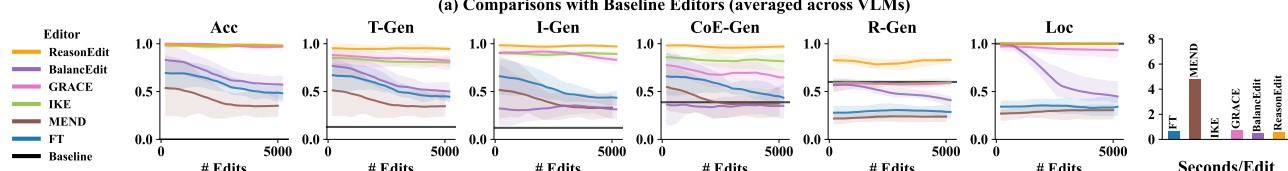

*Figure 4.* Sequential editing performance and efficiency across editors. ReasonEdit achieves the best sample generalities, rationale-informed generalities, as well as high reliability and locality. Trajectories are smoothed using moving average with a 5-step window.

| Noise ratio | LLaVA-1.5-7B | | | | | | InstructBLIP-7B | | | | | | Qwen3-VL-8B | | | | | | Qwen3-VL-4B | | | | | |
|---|---|---|---|---|---|---|---|---|---|---|---|---|---|---|---|---|---|---|---|---|---|---|---|---|
| | Acc | I-Gen | T-Gen | R-Gen | CoE-Gen | Loc | Acc | I-Gen | T-Gen | R-Gen | CoE-Gen | Loc | Acc | I-Gen | T-Gen | R-Gen | CoE-Gen | Loc | Acc | I-Gen | T-Gen | R-Gen | CoE-Gen | Loc |
| No noise | 1.00 | 0.97 | 0.97 | 0.86 | 0.98 | 1.00 | 0.98 | 0.95 | 0.96 | 0.87 | 0.96 | 1.00 | 0.98 | 0.94 | 0.92 | 0.80 | 0.95 | 1.00 | 0.99 | 0.91 | 0.91 | 0.80 | 0.96 | 1.00 |
| 10% | 0.99 | 0.98 | 0.97 | 0.83 | 0.99 | 1.00 | 0.99 | 0.95 | 0.96 | 0.84 | 0.97 | 1.00 | 0.98 | 0.94 | 0.91 | 0.78 | 0.98 | 1.00 | 0.95 | 0.93 | 0.91 | 0.81 | 0.97 | 1.00 |
| 30% | 0.99 | 0.98 | 0.97 | 0.83 | 0.98 | 1.00 | 0.98 | 0.95 | 0.96 | 0.83 | 0.98 | 1.00 | 0.99 | 0.94 | 0.92 | 0.78 | 0.97 | 1.00 | 0.96 | 0.93 | 0.92 | 0.80 | 0.97 | 1.00 |
| 50% | 0.99 | 0.97 | 0.98 | 0.82 | 0.99 | 1.00 | 0.99 | 0.95 | 0.96 | 0.83 | 0.97 | 1.00 | 0.99 | 0.94 | 0.92 | 0.78 | 0.97 | 1.00 | 0.96 | 0.93 | 0.92 | 0.81 | 0.97 | 1.00 |

(a) Inject noise into reasoning.

| Noise ratio | LLaVA-1.5-7B | | | | | | InstructBLIP-7B | | | | | | Qwen3-VL-8B | | | | | | Qwen3-VL-4B | | | | | |
|---|---|---|---|---|---|---|---|---|---|---|---|---|---|---|---|---|---|---|---|---|---|---|---|---|
| | Acc | I-Gen | T-Gen | R-Gen | CoE-Gen | Loc | Acc | I-Gen | T-Gen | R-Gen | CoE-Gen | Loc | Acc | I-Gen | T-Gen | R-Gen | CoE-Gen | Loc | Acc | I-Gen | T-Gen | R-Gen | CoE-Gen | Loc |
| No noise | 1.00 | 0.97 | 0.97 | 0.86 | 0.98 | 1.00 | 0.98 | 0.96 | 0.95 | 0.87 | 0.96 | 1.00 | 0.98 | 0.94 | 0.89 | 0.80 | 0.95 | 1.00 | 0.99 | 0.91 | 0.91 | 0.80 | 0.96 | 1.00 |
| 10% | 0.99 | 0.97 | 0.97 | 0.83 | 0.98 | 1.00 | 0.97 | 0.95 | 0.95 | 0.83 | 0.97 | 1.00 | 0.98 | 0.94 | 0.91 | 0.77 | 0.94 | 1.00 | 0.96 | 0.93 | 0.91 | 0.79 | 0.95 | 1.00 |
| 30% | 0.99 | 0.98 | 0.97 | 0.81 | 0.94 | 1.00 | 0.98 | 0.95 | 0.95 | 0.80 | 0.96 | 1.00 | 0.96 | 0.94 | 0.89 | 0.76 | 0.92 | 1.00 | 0.96 | 0.92 | 0.91 | 0.77 | 0.93 | 1.00 |
| 50% | 1.00 | 0.98 | 0.97 | 0.77 | 0.93 | 1.00 | 0.98 | 0.95 | 0.95 | 0.77 | 0.95 | 1.00 | 0.98 | 0.93 | 0.90 | 0.74 | 0.90 | 1.00 | 0.96 | 0.93 | 0.90 | 0.76 | 0.92 | 1.00 |

(b) Replace reasoning with noise.

*Table 2.* Robustness of ReasonEdit given noisy human reasoning. In (a), noisy statements are injected while preserving the original reasoning facts. In (b), a portion of the original reasoning facts is replaced with noise.

answer is directly provided in the prompt. The lower R-Gen and CoE-Gen are expected since no reasoning is provided. When using reasoning entries, we observe that ReasonEdit's performance approaches the ceiling accuracy where VLMs infer the answer from ground-truth human reasoning. These results show that reasoning entries are critical for reasoning-based generalization.

**(2) Poor image-/text-generality by biased single layer embeddings.** We then ablate our dual embedding method by comparing it with the best vision, language layers according to their best respective modularity. As shown in Fig. 5 (a), using a single vision layer to embed image–text inputs as codebook keys yields good text generality but *poor image generality*. This confirms vision layers have vision bias, where queries with perturbed images fall far from the target region and retrieve incorrect or irrelevant text. In contrast, picking a language layer yields better image generality but *low text generality*. This confirms language

layers have language bias, where queries tend to retrieve facts from irrelevant images that share high textual similarity with it. Finally, our dual embedding based on balanced vision-language modularity yields the best text and image generality, resolving single modality biases.

**(3) Influence of number of nearest neighbors $K$.** We next study how the number of retrieved values impacts ReasonEdit's performance. In Fig. 5 (b), we see that a small $K$ (e.g., $1, 3$) leads to the worst performance, because the neighbors must be most-accurate. Larger $K$ ($K \geq 5$) increases performance, and improvements largely taper above $K = 5$ for LLaVA and InstructBLIP, and $K = 7$ for Qwen3 models. This suggests that 5 is a reasonable default, though the optimal choice depends on the VLM and can be tuned via small-scale pre-editing experiments.

**(4) Sensitivity analysis on retrieval rejection threshold.** We next compare different thresholds for rejecting retrieval— If a query is not in the top $p$-th percentile of pairwise key distances, retrieval is rejected. In Fig. 5 (c), $p \leq 10$ limits retrieval and lowers R-Gen and CoE-Gen, while $p = 50$ enables more retrieval, improving generality while preserving locality. $p = 75$ mildly harms performance, indicating retrieval-based ReasonEdit is generally robust to over-retrieval, with 10 to 50 as good defaults.

**(5) Influence of key merging.** We finally examine the influence of merging keys by measuring the average memory overhead incurred when storing an edit in the codebook. Our algorithm successfully merges largely overlapping keys, reducing memory usage to around 80% of that without merging while preserving editing performances (see Appendix D.4).

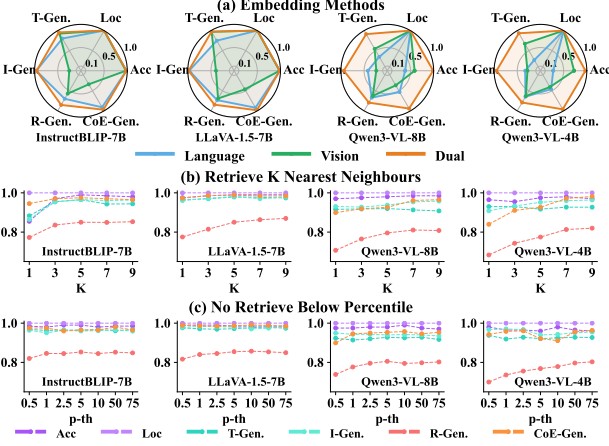

*Figure 5.* Results for ablation studies.

| Setting | LLaVA-1.5-7B | | | | | | InstructBLIP-7B | | | | | | Qwen3-VL-8B | | | | | | Qwen3-VL-4B | | | | | |
|---|---|---|---|---|---|---|---|---|---|---|---|---|---|---|---|---|---|---|---|---|---|---|---|---|
| | Acc | I-Gen | T-Gen | R-Gen | CoE-Gen | Loc | Acc | I-Gen | T-Gen | R-Gen | CoE-Gen | Loc | Acc | I-Gen | T-Gen | R-Gen | CoE-Gen | Loc | Acc | I-Gen | T-Gen | R-Gen | CoE-Gen | Loc |
| Prompt GT answer (ceiling) | 1.00 | 0.95 | 0.99 | 0.67 | 0.76 | 1.00 | 1.00 | 0.91 | 0.95 | 0.64 | 0.79 | 1.00 | 1.00 | 0.92 | 0.90 | 0.61 | 0.70 | 1.00 | 1.00 | 0.95 | 0.90 | 0.61 | 0.71 | 1.00 |
| Edit w/ answer entries only | 0.98 | 0.89 | 0.96 | 0.65 | 0.72 | 1.00 | 0.99 | 0.85 | 0.91 | 0.62 | 0.75 | 1.00 | 0.98 | 0.88 | 0.87 | 0.61 | 0.67 | 1.00 | 0.98 | 0.91 | 0.87 | 0.59 | 0.69 | 1.00 |
| Prompt GT reason (ceiling) | 0.85 | 0.84 | 0.80 | 0.92 | 0.89 | 1.00 | 0.83 | 0.86 | 0.78 | 0.94 | 0.83 | 1.00 | 0.77 | 0.80 | 0.75 | 0.89 | 0.80 | 1.00 | 0.79 | 0.81 | 0.77 | 0.91 | 0.82 | 1.00 |
| Edit w/ reason entries only | 0.81 | 0.81 | 0.80 | 0.87 | 0.82 | 1.00 | 0.80 | 0.74 | 0.75 | 0.87 | 0.81 | 1.00 | 0.76 | 0.76 | 0.73 | 0.84 | 0.78 | 1.00 | 0.76 | 0.77 | 0.75 | 0.83 | 0.79 | 1.00 |

*Table 3.* Isolating the effects of answer and reasoning entries in ReasonEdit. GT stands for ground truth.

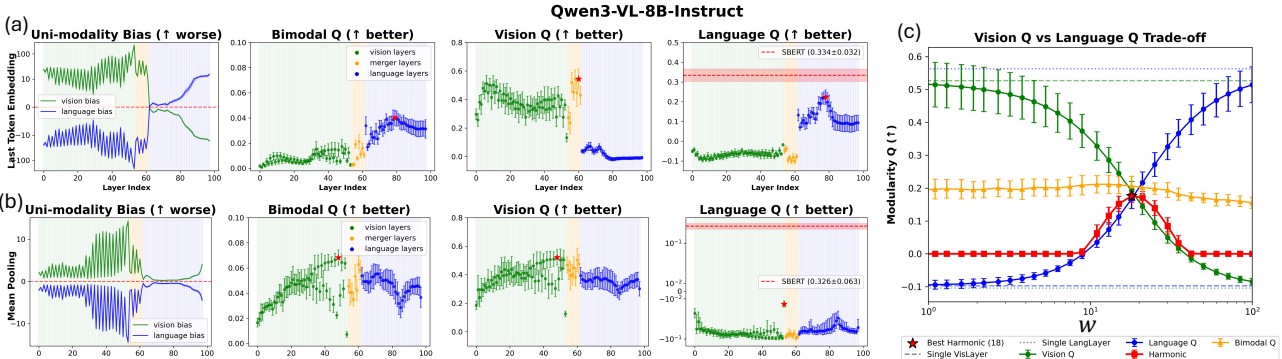

*Figure 6.* Network modularity provides a principled way to compare multimodal embedding methods, guiding (a) VLM layer selection, (b) the choice between last-token and mean-pooled embeddings, and (c) the choice of balancing weight $w$ in the dual embedding. The green, yellow, and blue backgrounds correspond to the vision, merger, and language blocks, respectively.

## 4.7. Topology-Aware Embedding Method Selection

Lastly, we study how network modularity provides a principled way to select multimodal embeddings to align with the desired topology, guiding (a) VLM layer selection, (b) the choice of last token or mean-pooled tokens, and (c) the choice of balancing weight $w$ in the dual embedding. Fig. 6 shows results for QWEN3-VL-8B-INSTRUCT . Similar results observed for other VLMs are in Appendix Fig. 7.

In panel (a), when using last-token embeddings, vision bias appears in vision-block (where green line in first plot is above 0), and is stronger in middle to later layers. Language bias appears in the language block (where blue line above 0), and is stronger in later layers. Correlated with biases, $\widehat{Q}_{\text{vis}}$ is higher in vision and merger blocks, while $\widehat{Q}_{\text{lang}}$ is higher in language layers, and $\widehat{Q}_{\text{bi}}$ peaks in language layers. These suggest last token embeddings are clustered predominantly by image in vision layers and by text in language layers.

In panel (b), using mean-pooled embeddings over all tokens, vision bias appears across all layers (though much less severe than using the last-token embedding), suggesting that mean pooling is dominated by image representations, likely because image tokens outnumber text tokens in the image–text prompt during aggregation. Similarly, $\widehat{Q}_{\text{vis}}$ and $\widehat{Q}_{\text{lang}}$ are highest at the respective blocks. However, $\widehat{Q}_{\text{bi}}$ peaks in latter vision layers and substantially exceeds the maximum achieved by last-token embeddings, suggesting that the embedding that aligns with the desired multimodal topology lies in the later vision layers using mean pooling. Higher $\widehat{Q}_{\text{bi}}$ and lower vision bias in later vision layers are similarly observed in other VLMs. This also motivates our use of mean-pooled embeddings from the vision layer that maximizes $\widehat{Q}_{\text{bi}}$ in the dual embedding. We provide an

ablation on using $Q_{vis}$ or $Q_{lang}$ to select $l$ in Appendix D.6.

In panel (c), over the balancing weight $w$, we observe a vision-language modularity trade-off in the dual embedding method. Embeddings transition from vision-biased to balanced then to language-biased topology as $w$ increases (see embedding network visualizations in Fig. 7). The $w$ in the dual embedding that results in more balanced multimodal topology can be selected by maximizing the harmonic mean of the two modularities, around which $\widehat{Q}_{\text{bi}}$ is also higher.

## 5. Conclusion

We introduce reasoning-enhanced VLM editing, where users provide detailed reasoning to guide editors in fixing errors made by VLMs on realistic, reasoning-heavy vision question answering tasks. We propose ReasonEdit, the first reasoning-enhanced VLM editor. To select the desired multimodal latent representation, we propose analyzing relationships between embeddings as a *graph*, and use modularity as a principled criterion to evaluate and select multimodal embeddings. ReasonEdit achieves state-of-the-art editing performance. Most importantly, it enables new forms of generalization: rationale generality and chain-of-error generality. It also improves image and text generality via a novel topology-balanced dual embedding method while preserving high locality and computational efficiency. Its reliable performance in sequential editing highlights practicality for real-time editing, where humans interact with a model and provide feedback when errors occur. As a new problem setup, promising future directions include editing errors in model-generated chains of thought through human interaction and intervention, and explicitly editing causal relationships among facts in the reasoning chain.

## Acknowledgements

This work was sponsored in part by NIH Award #1OT2OD038079-01. We thank University of Virginia's High Performance Computing team for computing resources. We also thank the reviewers for their time and constructive feedback.

## Impact Statement

This paper presents work with the goal to advance the field of Machine Learning. There are many potential societal consequences of our work, none of which we feel must be specifically highlighted here. There are several limitations of ReasonEdit. First, the search over the codebook at inference time becomes more expensive as the codebook grows larger, which is a limitation shared by all retrieval-based editors (e.g., GRACE, BalancEdit, IKE). Careful management of GPU computing resources can mitigate this limitation. Second, ReasonEdit assumes users provide accurate and structured reasoning. However, human reasoning can be noisy, incomplete, or incorrect. The current paper provides an initial evaluation of robustness under noisy reasoning, but a more comprehensive study remains an important direction for future work.

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

# A. Topology-Aware Multimodal Embedding

Prior retrieval-based editors such as the extended GRACE (Cheng et al., 2023; Guo et al., 2025) and BalancEdit (Guo et al., 2025) use a single-layer embedding of image-text pairs as keys, and retrieve the correct label for a question based on the Euclidean distance between a query (also an image–text pair) embedding and the keys. The decision to use a single VLM layer, and to adopt either the average of all token embeddings or the last-token embedding, was made post hoc by ablating editing performance (Cheng et al., 2023; Hartvigsen et al., 2023; Zeng et al., 2025; Guo et al., 2025). In this section, we quantify how well **an embedding method** $\mathcal{E}(\cdot)$ of bimodal image–text data aligns with the expected unimodal or bimodal topology, and reveal an interesting vision–language topology trade-off in VLMs. Inspired by network science, we introduce **sample modularity**, a metric to quantify topology alignment, and a **unimodal bias** metric to reveal whether an embedding method is biased toward a single modality.

## A.1. Sample Modularity

**Newman's Modularity** $Q$. In network science, *modularity* is a graph-theoretic measure of how well a network's connectivity aligns with a given partition (clustering) of its nodes (Newman, 2004; 2006). For a weighted network $G = (V, A)$ with node set $V$ and adjacency matrix $A$, let $A_{uv}$ denote the edge weight between nodes $u$ and $v$. Define the strength of node $u$ as $a_u = \sum_v A_{uv}$, and the total edge weight as $m = \frac{1}{2} \sum_{u,v} A_{uv}$. A partition $g : V \to \{1, \ldots, C\}$ assigns each node $u$ to a group $g(u)$. The modularity of this partition is

$$Q(A; g) = \frac{1}{2m} \sum_{u,v} \left( A_{uv} - \frac{a_u a_v}{2m} \right) \mathbf{1}[g(u) = g(v)]. \quad (1)$$

This score measures how much more edge weight lies *within* groups than would be expected by chance under a null model that preserves node strengths. Intuitively, a good partition is one in which edge weights (connectivity) are high within each group and low between groups. Larger $Q$ indicates stronger agreement between the network and the partition.

**Embedding Similarity Network Construction.** Let $i$ denote an image and $t$ denote a text. We write $z = \mathcal{E}(i, t) \in \mathbb{R}^d$ for the $d$-dimensional embedding of the image-text pair $(i, t)$. Each embedding method $\mathcal{E}(\cdot)$ induces a weighted similarity network $G = (V, A)$ over image-text pairs, where each node $z_u \in V$ corresponds to the embedding $\mathcal{E}(i, t)$ of a pair $(i, t)$, and each edge weight $A_{z_u z_v}$ is the similarity between nodes $z_u$ and $z_v$. We define the pairwise Euclidean distance and its normalized similarity as

$$d(u, v) = \|z_u - z_v\|_2, \qquad A_{uv} = \frac{d_{\max} - d(u, v)}{d_{\max} - d_{\min}},$$

where $d_{\min}$ and $d_{\max}$ are computed over all node pairs in the network and $A_{uu} = 0$. Note that this embedding similarity network with $A_{uv} \in [0, 1]$ is invariant to the scale of the embedding space $\mathbb{R}^d$, enabling direct comparison of modularity across VLM layers and embedding methods. The proof is provided in Appendix A.4

**Sample Modularity $\widehat{Q}$.** Constructing the full similarity graph over all possible image-text pairs in the entire embedding space is infeasible. Instead, we approximate its topology by Monte Carlo sampling. We first construct a *sample network* by randomly sampling $n$ image-text pairs $\{(i_k, t_k)\}_{k=1}^n$, treating the embedding vector of each pair as a node, and constructing an embedding similarity network $G_n = (V_n, A_n)$, where $V_n = \{\mathcal{E}(i_k, t_k)\}_{k=1}^n$ denotes the node set and $A_n$ is the weighted adjacency matrix. We compute the modularity of this sample network $A_n$ with respect to the expected partition denoted by $g_n$, using Eq. (1). Repeating this procedure $B$ times without replacement yields $B$ independent sample networks, each consisting of a random batch of $n$ image-text pairs. Lastly, we use the average modularity across the $B$ sample networks as a bootstrap estimator to obtain a *sample modularity*:

$$\widehat{Q} = \frac{1}{B} \sum_{b=1}^B Q\left(A_n^{(b)}; g_n^{(b)}\right), \tag{2}$$

Higher $\widehat{Q}$ indicates better alignment between the observed network $G_n$ and the target expected partition $g_n$, while negative values ($\widehat{Q} < 0$) indicate that nodes are more connected across clusters than within, suggesting misalignment with the target structure.

**Expected Unimodal and Bimodal Topology Via Partitioning.** We define the expected topology of a network through an *expected partition*, which specifies a clustering of nodes that reflects the desired semantic structure of an embedding method of bimodal data. Based on the expected partitions, we introduce three sample modularity metrics for image-text bimodal data.

**Definition A.1** (Expected Unimodal Partition)**.** The expected unimodal partition is a node partition in which nodes with high similarity with respect to *a single modality* are assigned to the same cluster, while nodes with low unimodal similarity are assigned to different clusters.

**Definition A.2** (Expected Bimodal Partition)**.** The expected bimodal partition is a node partition in which nodes with high similarity with respect to *both modalities* are assigned to the same cluster, while nodes with low similarity in either modality (or both) are assigned to different clusters.

**Vision Modularity $\widehat{Q}_{\text{vis}}$.** Given a random batch of $n$ image-text pairs, we construct the full cross-product of

batch images and texts: $V = \{\langle i_a, t_b \rangle : a, b \in \{1, \ldots, n\}\}$, where $i_a$ denotes the $a$-th image in the batch and $t_b$ denotes the $b$-th text. This yields $n^2$ nodes: for each original pair, there are $(n-1)$ nodes sharing the same image with different text, and $(n-1)$ nodes sharing the same text with different images. The vision partition groups nodes based on *image identity*, regardless of text, that is, all nodes sharing the same image should be assigned to the same cluster: $g_{\text{vis}}(\langle i_a, t_b \rangle) = a$. Let $A_{\text{vis}} \in \mathbb{R}^{n^2 \times n^2}$ denote the embedding similarity network of the $n^2$ nodes. The corresponding vision sample modularity is $\widehat{Q}_{\text{vis}} = \frac{1}{B} \sum_{b=1}^B Q(A_{\text{vis}}; g_{\text{vis}})$. A high $\widehat{Q}_{\text{vis}}$ reflects that the embedding method of the bimodal data aligns well with the expected image topology, where samples with similar images are embedded closer than those with different images.

**Language Modularity $\widehat{Q}_{\text{lang}}$.** We define language modularity in the same fashion, except the partition is based on *text identity* rather than image. Specifically, all nodes sharing the same text index are assigned to the same cluster: $g_{\text{lang}}(\langle i_a, t_b \rangle) = b$. The corresponding sample modularity is $\widehat{Q}_{\text{lang}} = \frac{1}{B} \sum_{b=1}^B Q(A_{\text{lang}}; g_{\text{lang}})$. A high $\widehat{Q}_{\text{lang}}$ reflects that the embedding method of the bimodal data aligns well with the expected text topology, where samples with similar text are embedded closer than those with different text.

**Bimodal Modularity $\widehat{Q}_{\text{bi}}$.** We define bimodal modularity in the same fashion as vision and language modularity, except the partition is based on *joint image-text identity*. For each anchor pair $(i_k, t_k)$ in a sampled batch of size $n$, we include two types of additional nodes: (1) all mismatched image-text pairs across the batch, including $n-1$ "same image, different text" combinations $\{(i_k, t_j) \mid j \neq k\}$ and $n-1$ "same text, different image" combinations $\{(i_j, t_k) \mid j \neq k\}$; (2) $2(n-1)$ augmented views $\{(i_k^{(r)}, t_k^{(r)})\}_{r=1}^{2(n-1)}$ per anchor, generated via image and text augmentation (details in Appendix A.6). The expected bimodal partition assigns each anchor and its augmentation variants to the same cluster: $g_{\text{bi}}(\langle i_k, t_k \rangle) = g_{\text{bi}}(\langle i_k^{(r)}, t_k^{(r)} \rangle) = k$. Let $A_{\text{bi}} \in \mathbb{R}^{|V^*| \times |V^*|}$ denote the adjacency matrix of the embedding similarity network constructed from all nodes in $V^*$. The corresponding sample modularity is $\widehat{Q}_{\text{bi}} = \frac{1}{B} \sum_{b=1}^B Q(A_{\text{bi}}; g_{\text{bi}})$. A high $\widehat{Q}_{\text{bi}}$ reflects that the embedding method of the bimodal data aligns well with the expected bimodal topology, where samples sharing both similar image and text are embedded closer in a cluster than mismatched combinations.

Note that sample modularity is computed on a randomly sampled batch of $n$ aligned image-text pairs, and repeating this procedure over $B$ independent batches serves as a Monte Carlo approximation of topology preservation. Increasing $n$ enlarges the node set (e.g., $|V| = n^2$ for vision/language modularity), which reduces estimator vari-

ance and yields more stable and contrastive modularity estimates. Detailed derivations are provided in Appendix A.5.

## A.2. Unimodal Biases

**Definition A.3** (Vision Bias). We define *vision bias* of an embedding method $\mathcal{E}(\cdot)$ as the tendency for the embedding to locate (same image, different text) pairs closer than (similar image, same text) pairs to a given (image, text) pair.

To quantify vision bias, for a given image-text pair $\langle i, t \rangle$, we induce two subgraphs: an image-perturbation subgraph $G_{\text{img}}^{+}$, formed by a set of (mildly augmented image, same text) pairs $\{\langle i^{+}, t \rangle\}$, and a text-mismatch subgraph $G_{\text{txt}}^{-}$, formed by a set of (same image, randomly sampled mismatched text) pairs $\{\langle i, t^{-} \rangle\}$. Let $\overline{w}(G)$ denote the average edge weight (embedding distance) of subgraph $G$. The mismatched texts are sampled from a pool of 500 diverse common fact sentences. We compute vision bias as

$$\text{Bias}_{\text{vis}}(\langle i, t \rangle) = \overline{w}\left(G_{\text{img}}^{+}\right) - \overline{w}\left(G_{\text{txt}}^{-}\right).$$

To obtain a stable estimate, we compute $\text{vis\_bias}(\langle i, t \rangle)$ over a randomly sampled batch of $n$ image-text pairs and average the resulting values. Repeating this procedure over $B$ independent batches yields a Monte Carlo estimate of vision bias, reported as the mean (and standard deviation) across all samples. For embedding-distance-based editors, such a bias leads to poor image generality, as it is overly sensitive to perturbations of the image and pushes similar images paired with the same text to distant regions in the embedding space. It can also result in poor locality, as irrelevant text is more likely to be retrieved purely due to the image similarity in bimodal data.

**Definition A.4** (Language Bias). Similarly, we define *language bias* of $\mathcal{E}(\cdot)$ as the tendency for the embedding to locate (different image, same text) pairs closer than (same image, similar text) pairs to a given (image, text) pair.

To quantify language bias, for a given image-text pair $\langle i, t \rangle$, we induce two subgraphs: a text-perturbation subgraph $G_{\text{txt}}^{+}$, formed by a set of (same image, mildly perturbed text) pairs $\{\langle i, t^{+} \rangle\}$, and an image-mismatch subgraph $G_{\text{img}}^{-}$, formed by a set of (randomly sampled mismatched image, same text) pairs $\{\langle i^{-}, t \rangle\}$. Let $\overline{w}(G)$ denote the average edge weight (embedding distance) of subgraph $G$. The mismatched images are sampled from (Lorem Picsum Contributors, 2026). We compute language bias as

$$\text{Bias}_{\text{lang}}(\langle i, t \rangle) = \overline{w}\left(G_{\text{txt}}^{+}\right) - \overline{w}\left(G_{\text{img}}^{-}\right).$$

To obtain a stable estimate, we compute $\text{lang\_bias}(\langle i, t \rangle)$ over a randomly sampled batch of $n$ image-text pairs and average the resulting values. Repeating this procedure over $B$ independent batches yields a Monte Carlo estimate of language bias, reported as the mean (and standard deviation) across all samples. For embedding-distance-based editors, such a bias leads to poor text generality, as it is overly sensitive to perturbations of the text and pushes similar texts paired with the same image to distant regions in the embedding space. It can also result in poor locality, as irrelevant content from different images is more likely to be retrieved purely due to text similarity in bimodal data.

## A.3. Data-agnostic Embedding Selection Strategy

We compare the topology-aware embedding selection's hyperparameter across three different image sources: COCO(Chen et al., 2015), ImageNet(Russakovsky et al., 2015), and Flickr30k(Plummer et al., 2015). The selected vision layer index $l$ and balancing weight $w$ for each VLM are shown below.

*Table 4.* Selected hyperparameters across different image sources

| vision layer $l$ index | LLaVA | InstructBLIP | Qwen4B | Qwen8B |
|---|---|---|---|---|
| COCO | 19 | 38 | 21 | 24 |
| ImageNet | 19 | 38 | 20 | 24 |
| Flickr30k (new) | 19 | 38 | 20 | 23 |
| **balancing weight $w$** | **LLaVA** | **InstructBLIP** | **Qwen4B** | **Qwen8B** |
| COCO | 7 | 8 | 41 | 18 |
| ImageNet | 8 | 8 | 40 | 18 |
| Flickr30k (new) | 7 | 8 | 40 | 19 |

As shown above, the selected hyperparameters are highly consistent across the three image sources. The chosen vision layer indices remain identical or differ by at most one layer, and the balancing weights are also very close. This strong consistency indicates that our topology-aware embedding selection is largely data-agnostic.

## A.4. Scale-invariant Sample Modularity

**Embedding-scale invariance.** We first prove that the constructed similarity network $A$ is invariant to global embedding rescaling. If embeddings are rescaled by a constant $\alpha > 0$, i.e., $z'_u = \alpha z_u$, then distances satisfy $d'(u, v) = \|z'_u - z'_v\|_2 = \alpha \|z_u - z_v\|_2 = \alpha d(u, v)$. Consequently, $d'_{\min} = \alpha d_{\min}$ and $d'_{\max} = \alpha d_{\max}$, and the normalized similarity remains unchanged:

$$A'_{uv} = \frac{d'_{\max} - d'(u, v)}{d'_{\max} - d'_{\min}} = \frac{\alpha d_{\max} - \alpha d(u, v)}{\alpha d_{\max} - \alpha d_{\min}} = A_{uv}.$$

**Weight-scale invariance of Newman's modularity.** Newman's modularity is invariant to uniform scaling of all edge weights. Let $A' = cA$ for $c > 0$. Then $k'_u = \sum_v A'_{uv} =$

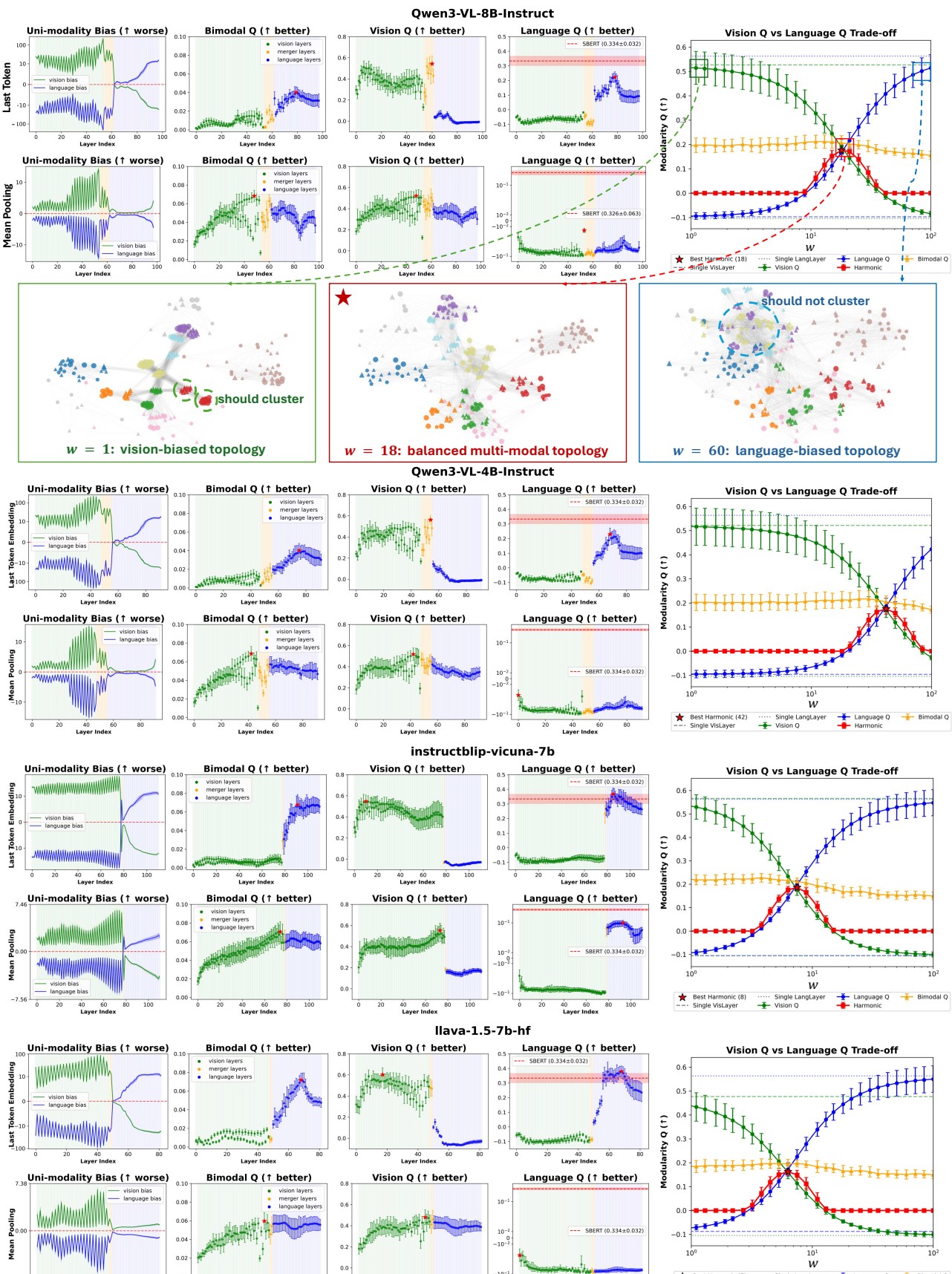

*Figure 7.* Vision bias, language bias, vision sample modularity ($Q_{\text{vis}}$), language modularity ($Q_{\text{lang}}$), and bimodal modularity ($Q_{\text{bi}}$) across layers of four VLMs, and the vision–language topology trade-off controlled by the text-scaling factor $w$ in the dual embedding. The green, yellow, and blue backgrounds correspond to the vision, merger, and language blocks, respectively. Example embedding networks for QWEN3-VL-8B-INSTRUCT at varying $w$ visualize the shift in topology alignments. Each network shows 10 edits (large circles) and their patches (small circles) and augmentations (large/small triangles).

$ck_u$ and $m' = \frac{1}{2}\sum_{u,v} A'_{uv} = cm$. The null-model term scales as

$$\frac{k'_u k'_v}{2m'} = \frac{(ck_u)(ck_v)}{2(cm)} = c\frac{k_u k_v}{2m}.$$

Substituting into Eq. (1) yields

$$Q(A';g) = \frac{1}{2cm}\sum_{u,v} c\left(A_{uv} - \frac{k_u k_v}{2m}\right)\mathbf{1}[g(u) = g(v)]$$

$$= Q(A;g).$$

Therefore, these scale-invariance properties allow us to directly compare modularity scores across embeddings extracted from different VLM layers and across different embedding methods.

### A.5. Effect of number of nodes $n$ in a sample network

Sample modularity is computed on a randomly sampled batch of $n$ aligned image–text pairs. Repeating this over $B$ independent batches provides a Monte Carlo estimate of topology preservation. Increasing $n$ enlarges the node set (e.g., $|V| = n^2$ for vision or language modularity), which reduces estimator variance and yields more stable values. Importantly, while the absolute modularity scores change with $n$, the *relative ranking* of embedding strategies and VLM layers is preserved.

In the unimodal (vision/language) case, the target partition contains $n$ clusters of size $n$, and the fraction of within-cluster node pairs is

$$\frac{n\binom{n}{2}}{\binom{n^2}{2}} = \frac{n-1}{n^2-1} = \frac{1}{n+1} \approx \frac{1}{n}.$$

As $n$ increases, the proportion of cross-cluster edges grows, sharpening the modularity contrast and reducing noise. This improves the stability of estimates but does not alter the ranking of embedding representations.

In the bimodal case, we set the number of augmentations per anchor to $2(n-1)$, yielding

$$|V_{\text{aug}}| = 2n(n-1), \quad |V_{\text{swap}}| = 2n(n-1),$$

and a total node set

$$|V^*| = n + 4n(n-1).$$

This balanced construction scales with $n$ while preserving the same anchor–positive–negative structure. As a result, increasing $n$ mainly reduces variance and tightens confidence intervals; the ordering of VLM layers and embedding strategies remains consistent across $n$.

### A.6. Augmentation

We use augmentation in the bimodal modularity calculation and the key merging algorithm to construct semantically equivalent samples for a given image–text pair. Our augmentation procedure is as follows. We apply lightweight multimodal augmentation that stochastically perturbs images while minimally rewriting the associated text. For images, we pad the original with a mosaic background so it occupies $\sim 90\%$ of the final canvas, placing it at a random location and filling the remaining area with a $4 \times 4$ grid of tiles sampled from natural images downloaded from (Lorem Picsum Contributors, 2026), together with random noise (see example in Fig. 10). By using random images and noise to occupy the remaining 10% area around the original image, we aim to approximate an image that preserves 90% of the semantic meaning of the original image. Mosaic augmentation is a widely used strategy in image augmentation for vision machine learning (Zhang et al., 2024a; Dulal et al., 2022; Bochkovskiy et al., 2020). Then, we apply standard photometric and geometric jitter, including random rotation $\pm 10°$ and mild color jitter. For text, we prompt a small instruction-tuned LLM, Qwen2.5-1.5B-Instruct, with: "Rephrase this sentence while keeping the same meaning. Only output the rephrased sentence, nothing else." We sample at temperature $= 1.5$ (max 64 new tokens) to generate diverse English paraphrases.

## B. Evaluation Details

For a given edit $(i, t, y^+, r^+)$, where $i$ is the image, $t$ is the text prompt question, $y^+$ is the correct answer, and $r^+$ is the human reasoning containing human CoT sentences, we evaluate the correctness of the updated VLM on correcting related variants of this edit, while preserving the answers on the unrelated samples.

**Reliability**  The updated model should output the target answers on the edit set correctly.

$$\text{reliability} = \mathbb{E}_{(i,t,y^+,r^+)\in D_{\text{edit}}}\mathbf{1}\{f_{\text{new}}(i,t) = y^+\}$$

**Locality**  The updated model should retain the original predictions on the unrelated samples.

$$\text{Locality} = \mathbb{E}_{(i',t')\in U_{i,t}}\mathbf{1}\{f_{\text{new}}(i',t') = f_{\text{old}}(i',t')\}$$

Since we evaluate editors on real errors made by the VLM on each dataset, our locality evaluation is equivalent to measuring whether the updated model remains correct on samples that were originally correct. To prepare the unrelated set, rather than sampling unrelated samples uniformly at random (Cheng et al., 2023)—which may include points far from the edit distribution—we include two subsets from the correct set: (i) 100 randomly sampled correct samples,

and (ii) 100 correct samples whose questions are highly semantically similar to the edits, selected by choosing the top three most similar questions based on word overlap.

**Text Generality**  Let $R(t)$ denote the set of rephrased versions of question $t$. The updated model should output the correct answer when given any rephrased question.

$$\text{T-Gen} = \mathbb{E}_{(i,t,y^+,r^+)\in D_{\text{edit}}}\mathbf{1}\{f_{\text{new}}(i, R(t)) = y^+\}$$

For text generality evaluation dataset, we generate question variants using a rephrasing method, following Guo et al. (2025). We use GPT-4o to generate rephrased questions with the prompt: `Please rephrase the following question in 10 different ways: {question}.`

**Image Generality**  Let $R(i)$ denote similar images to $i$, the updated model should output the correct answer given similar images.

$$\text{I-Gen} = \mathbb{E}_{(i,t,y^+,r^+)\in D_{\text{edit}}}\mathbf{1}\{f_{\text{new}}(R(i), t) = y^+\}$$

For image generality evaluation data, we follow the process in prior work (Guo et al., 2025; Cheng et al., 2023). Our goal is to generate semantically similar images to evaluate whether the edited model still answers the question correctly under visual substitutions. We generate these images in two steps. (1) We retrieve captions for the raw images in A-OKVQA (Schwenk et al., 2022) and FVQA (Wang et al., 2017). A-OKVQA contains 17,656 unique COCO-2017 images(Chen et al., 2015). FVQA contains 1,332 unique COCO-2014 images and 858 unique ImageNet-2012 images (Russakovsky et al., 2015; Chen et al., 2015). We retrieve caption annotations for all COCO-sourced images in these datasets. For ImageNet-2012 images where captions are unavailable, we prompt Qwen3-8B (Yang et al., 2025) with `Describe this image in a short sentence.'` to obtain captions. We append the human reasoning that contains the visual details needed to answer the question to the end of each caption, to preserve details that are commonly omitted in the original caption. (2) We use two SOTA text-to-image diffusion models (Stable Diffusion 3 (Esser et al., 2024) and FLUX (Batifol et al., 2025)) to generate four images per caption, with two images per model, to capture model-selection variability in image generation.

**Rationale Generality**  The updated model should correctly answer questions about any intermediate facts provided in the human reasoning $r^+ = \{s_1, s_2, \ldots, s_{N_s}\}$. Let $S \subseteq r^+$ be any non-empty subset of reasoning facts, and let $(i_S, t_S)$ denote a new image–question pair that relies on $S$, with correct answer $y_S^+$. We define *rationale generality* as

$$\text{R-Gen} = \mathbb{E}_{(i,t,y^+,r^+)\in D_{\text{edit}},\, S\subseteq r^+}\mathbf{1}\{f_{\text{new}}(i_S, t_S) = y_S^+\}.$$

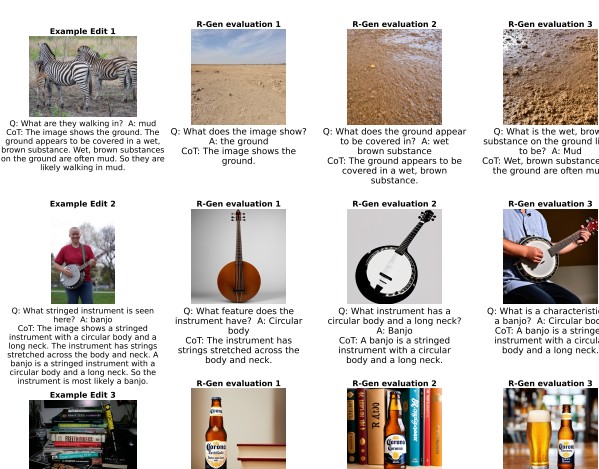

*Figure 8.* R-Gen Examples. For a given edit, each evaluation sample poses a question about a subset of facts in the human chain of thoughts. The updated VLM should answer correctly for images and questions about such intermediate facts.

To evaluate this, we construct the rationale generality evaluation dataset in two steps.

(1) *QA Synthesis.* We enumerate all $2^n - 1$ ordered subsets of the $n$ CoT sentences $\{s_1, s_2, \ldots, s_n\}$. For each subset, we prompt GPT-4o to generate a multiple-choice question:

```
You are creating a multiple-choice question
from the given facts: {subset}. Task:
(1) Write one question that can be answered
using only these facts. (2) Write one
correct answer. (3) Write three incorrect
but plausible answers.
```

(2) *Image Generation.* For each subset of facts, we generate a corresponding image using the text-to-image diffusion models, following the same image generation procedure in the image generality. Fig. 8 shows examples of the R-Gen evaluation dataset.

**Chain-of-Error (CoE) Generality**  For a given edit, let $\text{CoE} = \{e_1, e_2, \ldots\}$ denote the subset of chain-of-thought sentences that the VLM fails to recognize (**error-inducing**), and let $\{c_1, c_2, \ldots\}$ denote the remaining sentences that the VLM correctly recognizes. Chain-of-error generality measures whether the updated VLM can produce the correct answer $y^+$ to the original question $t$ when given new images $i_{\text{coe}}$ in which the error-inducing facts $\{e_1, e_2, \ldots\}$ are visually present, while the other conditions $\{c_1, c_2, \ldots\}$ vary. That is, the updated VLM should answer the question correctly when a new sample shares the same failure modes.

$$\text{CoE-Gen} = \mathbb{E}_{(i,t,y^+,r^+)\in D_{\text{edit}}}\mathbf{1}\{f_{\text{new}}(i_{\text{coe}}, t) = y^+\}$$

(1) *CoE Generation.* We enumerate all $2^n - 1$ ordered subsets of the $n$ CoT sentences $\{s_1, s_2, \ldots, s_n\}$ and verify each subset against the image using the VLM as

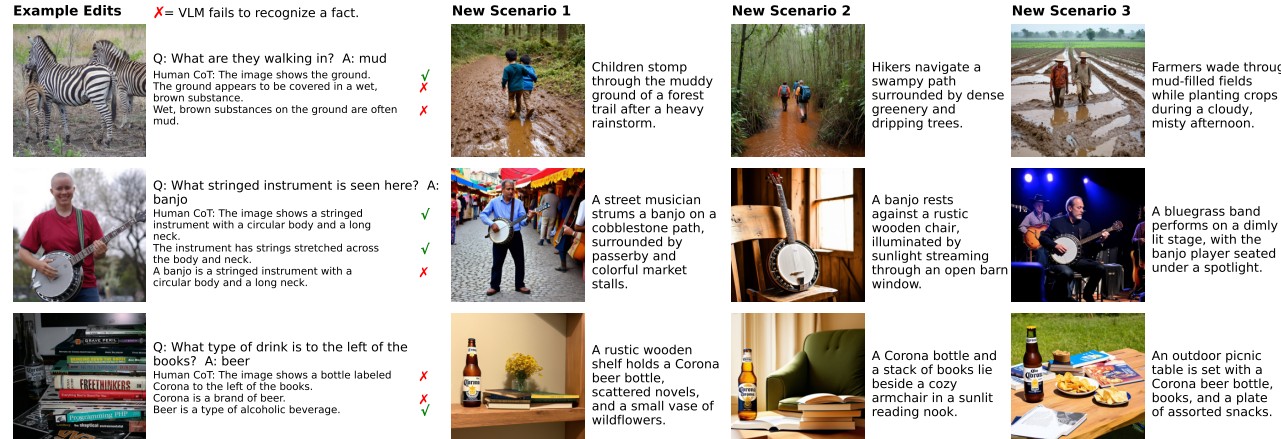

*Figure 9.* CoE-generality Examples. In the first example edit, the VLM fails to recognize (i.e., incorrectly answers "no" to) the second and third fact sentences in the ground-truth chain of thought, resulting in an error-inducing chain consisting of two sentences. Three new scenarios are then generated and paired with this error chain for the given edit and VLM.

a binary classifier. Specifically, we prompt the model: `Given the image, is the following statement correct? Statement:'{sentence_subset}'`. We compute $P$('Yes') and $P$('No') as follows. Given a prompt $x$ and image $I$, we evaluate the negative log-likelihood (NLL) for a single-token response $y \in \{$'Yes', 'No'$\}$:

$$\text{NLL}(y \mid I, x) = -\log P(y \mid I, x).$$

We then normalize over the two candidates with a softmax:

$$P(y) = \frac{e^{-\text{NLL}(y|I,x)}}{\sum_{y' \in \{\text{'Yes','No'}\}} e^{-\text{NLL}(y'|I,x)}}.$$

This yields a binary probability indicating whether the VLM fails to recognize a fact or a set of facts in the ground truth CoT. Then, a subset is labeled as *error-inducing* if $P$('No') $> 0.5$. Lastly, all error-inducing subsets are then merged by taking the union of their sentences to form a **error chain** (red-crossed facts in Fig. 9) for each edit.

(2) *Varying Other Visual Conditions.* Given the error chain, we use GPT-4o (temperature = 1.0) to generate three creative visual scenarios that can plausibly co-occur with the error-inducing facts in the error chain:

```
Given these visual facts: {error_chain},
generate 3 different creative scenarios
where ALL these facts would be visually
true. Requirements: Each scenario must be
exactly one sentence (less than 20 words).
Be creative but plausible. Describe what
would be visible in the image. Do not
contradict the given facts.
```

(3) *Image Generation.* For each generated scenario, we synthesize new images using Stable Diffusion 3 (Esser et al., 2024), following the same procedure as in image generality. The text-to-image prompt *prepends* each new scenario to the corresponding error-chain sentences as the full image description. Then, the diffusion model generates three

new images per edit using the description that preserve the same chain of errors while varying other visual conditions. Examples are shown in Fig. 9.

**Image Generation Quality Evaluation** We follow prior work (Cheng et al., 2023; Zhang et al., 2025a; Guo et al., 2025) to generate images for several evaluation metrics. Nonetheless, we perform a quality check on each generated image and regenerate images until they pass the following criteria. We verify each image using VQA-based validation with QWEN3-VL-8B-INSTRUCT : for a generated image, we ask the binary question "Does this image show: {text}?", where *text* is the prompt used to generate the image. We then compare the model's negative log-likelihoods for answering "yes" versus "no". Images with low verification confidence ($P(\text{yes}) < P(\text{no})$) are added to a regeneration list. We repeat this generation–verification process up to three times and discard any images that still fail after three attempts, which affects about $0.01\%$ of evaluation set. Specifically, this filtering removes 389 generated images for A-OKVQA and 74 for FVQA after up to three regeneration attempts.

## C. ReasonEdit Details

**Visual Evidence Patchification.** In ReasonEdit, users are allowed to provide cropped image patches as visual evidence for statements included in their reasoning. For example, in Fig. 1, a dermatologist can crop the lesion area to support the statement the lesion has an irregular shape," and crop the surrounding skin region to support the statement the patient has white skin color." In cases where no manual evidence patch is provided, we employ an automatic procedure that identifies image regions most relevant to a given sentence. Given an image $i$ and a sentence $s_j \in r^+$, we first apply a grid over the image to generate candidate patches at multiple spatial scales, capturing both fine-grained details

**An edit = (image, question, correct answer, human reasoning)**

Original (3×3 grid)     Augmented (mosaic)

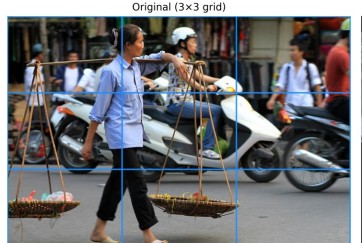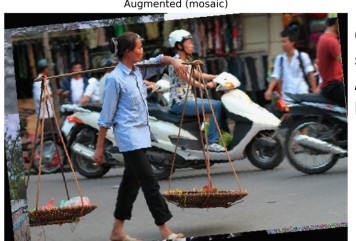

Q: What material is used to make the stick on the woman's shoulder?
A: bamboo
Human Reasoning:
  s1: The image shows a woman with a stick on her shoulder.
  s2: The stick appears firm but flexible.
  s3: Bamboo is a type of wood that is firm and flexible.
  s4: Bamboo grows commonly in Asian regions.

**Visual evidence patch selection for sentence
s = "Bamboo is a type of wood that is firm and flexible."**

**All (image patch, sentence) pairs**

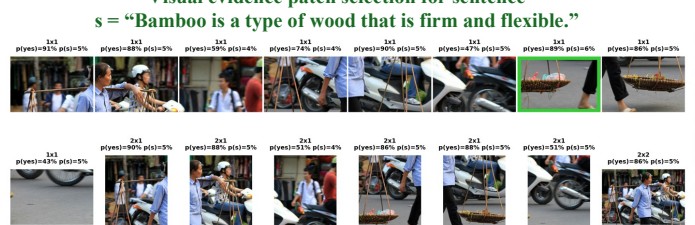

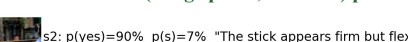
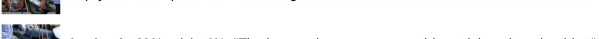
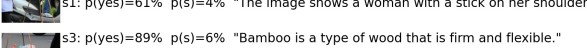

s2: p(yes)=90%  p(s)=7%  "The stick appears firm but flexible."

s1: p(yes)=53%  p(s)=4%  "The image shows a woman with a stick on her shoulder."

s1: p(yes)=61%  p(s)=4%  "The image shows a woman with a stick on her shoulder."

s3: p(yes)=89%  p(s)=6%  "Bamboo is a type of wood that is firm and flexible."

s4: p(yes)=92%  p(s)=7%  "Bamboo grows commonly in Asian regions."

s1: p(yes)=85%  p(s)=9%  "The image shows a woman with a stick on her shoulder."

s4: p(yes)=89%  p(s)=7%  "Bamboo grows commonly in Asian regions."

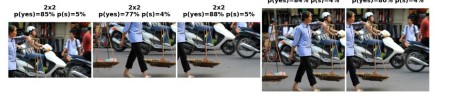

*Figure 10.* Visual Evidence Patchification Example.

and broader semantic context. For each patch, the VLM is prompted with the question "Does the image show $s_j$?". We retain patches for which the predicted probability of "yes" exceeds that of "no", and discard the rest. For the kept patches, we score relevance by computing the log-likelihood of the sentence $s_j$ conditioned on the patch $p$, based on the VLM's generation under the prompt "Describe this image." Lastly, we retain the highest-likelihood patch across all candidates as the strongest visual evidence, together with the highest-likelihood patch among the smallest patches to capture the most localized, fine-grained visual evidence. A patchification example is shown in Fig. 10.

To evaluate the quality of the automatic patchification process, we compute CLIP similarity between each automatically extracted patch and its associated reasoning text using the edit set from each VLM. Compared to the baseline CLIP similarity between the raw images and their ground-truth captions, our automatic patches achieve similar CLIP similarity.

*Table 5.* CLIP similarity between reasoning sentences and images

| CLIP similarity | LLaVA | InstructBLIP | Qwen4B | Qwen8B |
|---|---|---|---|---|
| Raw Images (baseline) | 0.279 (0.041) | 0.260 (0.042) | 0.273 (0.038) | 0.261 (0.043) |
| Auto Patches | 0.277 (0.036) | 0.267 (0.036) | 0.274 (0.037) | 0.273 (0.036) |

**Merging Keys.** When inserting a new key into the codebook, ReasonEdit applies a key merging procedure to prevent redundant entries while preserving semantic coverage.

Given a newly generated key with embedding $z$, we first estimate its local embedding-space radius. To do so, we construct augmented image–text pairs from the original edit by applying a mild image augmentation with mosaic padding (example in Fig. 10) and text paraphrasing (see Appendix A.6). We then compute the $\ell_2$ distances between the embedding of the original pair and the embeddings of its augmented variants, and set the local radius $r$ based on these distances. This radius characterizes the variability of the key under semantic-preserving perturbations. Next, we compare the new key against existing codebook entries. For each existing key with embedding $z_k$ and radius $r_k$, we detect overlap between $(z_k, r_k)$ and $(z, r)$ if both of the following conditions are satisfied: (1) the intersection area divided by the total area exceeds $90\%$ for both radius-defined circles, and (2) the embedding distance between the two keys is smaller than $10\%$ of both radii. If no overlap is detected, the new entry is added to the codebook independently. Otherwise, a merge occurs, in which we update the existing key by setting its embedding to the average of $z_k$ and $z$, and update its value by taking the union of the associated sentences from both entries. This merging procedure allows ReasonEdit to consolidate semantically redundant keys while maintaining a compact codebook that preserves coverage of previously observed corrections and reasoning facts.

# D. Experiment Details

## D.1. Edits Generation

In our Rationale-VQA datasets (A-OKVQA and FVQA), we let $D_{\text{edit}}$ denote the set of real errors made by the VLM (the error set), and $U_{i,t}$ denote the unrelated samples that the VLM answered correctly (the correct set). Because our VLMs generate free-form reasoning rather than short labels, the text-comparison-based label-flipping accuracy used in prior work (Guo et al., 2025; Hartvigsen et al., 2023; Cheng et al., 2023) becomes brittle and unreliable due to the mismatch between long explanations and concise targets. Moreover, it is not reasonable to force the updated model to output only a label ($y^+$), as doing so suppresses the reasoning capabilities these VLMs are designed to exhibit. Therefore, we prompt VLMs with regular image–question inputs, but determine the final answer by computing the highest softmax probability over the four candidate answers in the multiple-choice setting, following prior rationale VQA benchmark work (Schwenk et al., 2022; Wang et al., 2017). This serves as a necessary condition for a VLM to generate the correct answer, since a higher probability indicates the answer is more likely to appear in the generated text. An additional benefit of this prediction method is that it guarantees a valid answer is always selected, since our goal is to maintain stable and fair comparisons across all editors.

## D.2. Edited VLMs

QWEN3-VL-4B-INSTRUCT and QWEN3-VL-8B-INSTRUCT (Yang et al., 2025) embed visual tokens directly into a transformer decoder and fuse multi-level visual features with text representations for unified vision–language reasoning and generation. Model INSTRUCTBLIP-7B (Dai et al., 2023) is an instruction-tuned BLIP-2 variant that employs a frozen image encoder and a Q-Former to extract visual queries, which are then projected into a Vicuna-7B large language model for generation. LLAVA-1.5-7B (Liu et al., 2023) connects a pretrained CLIP ViT-L/14 vision encoder to a Vicuna/LLaMA-style decoder via a lightweight projection layer, and is trained with a two-stage alignment and instruction-tuning pipeline.

## D.3. Baseline Editors Selection

We compare ReasonEdit with five state-of-the-art editors adapted or proposed for VLM editing. 1) We finetune (FT) chosen layers of the edited model for given edits. This approach is the easiest to implement, but easily overfits during editing. 2) MEND (Mitchell et al., 2021) is a meta-learning based editor that trains a meta-network to predict new weights for the pretrained model for future edits. 3) In-context Knowledge Editing (IKE) (Qi et al., 2024) retrieves and prepends relevant facts to new prompts, and is adapted

in prior VLM editing work (Guo et al., 2025) to use an unsupervised retriever over in-distribution data to select facts from any previous edits. 4) GRACE (Hartvigsen et al., 2023) is a lifelong editor that stores edits as key–value mappings in a codebook, applies them only to inputs near past errors, and uses the last token embedding as keys. GRACE leaves the edited model's weights frozen. 5) BalancEdit (Guo et al., 2025) is specifically a VLM editor, and learns a dynamic influence radius for each edit. It finetunes a selected layer per edit and applies updated weights only when an input falls within an edit's radius. Beyond using each method as originally proposed, we also augment each to be reasoning-enabled (denoted editor-COT) to ensure editors receive the same inputs as ReasonEdit. For weight-updating editors (FT, MEND, BalancEdit), we train the VLM or editor to generate the correct answer followed by the human reasoning for each image–question edit. For retrieval-based editor GRACE, we populate its codebook with key–value entries for all factual sentences in the human reasoning, in addition to the entry for the correct answer. Besides these editors, we attempt to implement more state-of-the-art VLM editors. However, the implementation of MSCKE editor is not yet publicly available. LiveEdit requires large-scale pretraining on a complete edit dataset where each edit is paired with generality and locality samples. This is unsuitable for our editing settings where only human correction and reasoning are available for each edit.

## D.4. Implementation Details

For ReasonEdit, we concatenate a pretrained sentence encoder (`paraphrase-mpnet-base-v2` (Reimers & Gurevych, 2019)) with the VLM's vision layer selected by maximizing bimodal sample modularity in the dual embedding method. Sample modularity is estimated via Monte Carlo over $B$ independent sample networks. We set $B = 10$ because it is sufficient for variance to converge with reasonable computational cost. Each is constructed from a random batch of $n$ image–text pairs. Pairs are sampled from the combined pool of the two datasets, covering diverse images from COCO, and ImageNet (Chen et al., 2015; Russakovsky et al., 2015). We try different values of $n = 5, 10, 20$ (Supplemental Fig. 13) and pick $n = 10$ for both lower Monte Carlo variance and lower computational cost. The effect of $n$ on sample modularity is discussed in Appendix A.5. At inference, we set $K = 5$ for nearest-neighbor retrieval and use a no-retrieval percentile threshold of $p = 50\%$ across all VLMs. Ablation studies on them are presented in Sec. 4.6. To fairly compare ReasonEdit with prior editors, we keep each editor's configuration as close to their implementations as possible. We use the Adam optimizer with 100-500 steps and early stopping with a patience of 50 steps for all prior editors. We apply cosine learning-rate decay, with initial learning rates of $1\mathrm{e}{-3}$ for FT and BalancEdit and $1\mathrm{e}{-2}$ for

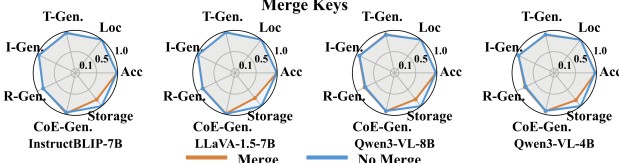

*Figure 11.* Merging keys reduces memory overhead while preserving the editing performance.

GRACE and MEND. We follow Guo et al. (2025) and edit the final MLP block in the language backbone for all editors.

## D.5. Influence of Key Merging

We examine the influence of merging keys by measuring the average memory overhead incurred when storing an edit in the codebook. Overlapping occurs when a user provides highly similar facts for the same image within an edit, or when repeated or similar facts are introduced across different edits. As shown in Fig. 11 and Table 6. Our algorithm successfully merges largely overlapping keys, as indicated in the lower-right corner of the radar plot: it reduces memory usage to about 80% of that without merging while preserving editing performances.

*Table 6.* Storage Before vs After Key Merging

| VLM | No Merge (KB/edit) | With Merge (KB/edit) | Reduction (%) |
|---|---|---|---|
| InstructBLIP-7B | 34.3 | 27.6 | 19.4% |
| LLaVA-1.5-7B | 31.7 | 24.1 | 24.2% |
| Qwen3-VL-8B | 32.8 | 26.1 | 20.3% |
| Qwen3-VL-4B | 28.7 | 23.6 | 17.9% |

## D.6. Selecting vision layer $l$ by $Q_{vis}$ vs. $Q_{lang}$.

Lastly, we ablate the criterion used to select the vision layer $l$ in the dual embedding. The reason for using $Q_{bi}$ is that it identifies a layer with more balanced topology among the already vision-biased vision layers. As shown in Table 7, selecting $l$ by $Q_{vis}$ yields performance close to $Q_{bi}$, while selecting $l$ by $Q_{lang}$ leads to substantially worse performance, especially on T-Gen. This is expected, since maximizing $Q_{lang}$ makes the dual embedding strongly text-biased.

## D.7. Supplemental Figures

| Criterion | LLaVA-1.5-7B | | | | | InstructBLIP-7B | | | | | Qwen3-VL-8B | | | | | Qwen3-VL-4B | | | | |
|---|---|---|---|---|---|---|---|---|---|---|---|---|---|---|---|---|---|---|---|---|
| | Acc | T-Gen | I-Gen | R-Gen | CoE-Gen | Acc | T-Gen | I-Gen | R-Gen | CoE-Gen | Acc | T-Gen | I-Gen | R-Gen | CoE-Gen | Acc | T-Gen | I-Gen | R-Gen | CoE-Gen |
| $Q_{bi}$ | 1.00 | 0.97 | 0.97 | 0.86 | 0.98 | 0.98 | 0.96 | 0.95 | 0.87 | 0.96 | 0.98 | 0.89 | 0.94 | 0.80 | 0.95 | 0.99 | 0.91 | 0.91 | 0.80 | 0.96 |
| $Q_{vis}$ | 0.99 | 0.96 | 0.93 | 0.84 | 0.95 | 0.96 | 0.95 | 0.91 | 0.85 | 0.95 | 0.97 | 0.88 | 0.91 | 0.78 | 0.94 | 0.98 | 0.90 | 0.89 | 0.78 | 0.91 |
| $Q_{lang}$ | 0.76 | 0.78 | 0.80 | 0.72 | 0.74 | 0.73 | 0.74 | 0.79 | 0.70 | 0.79 | 0.57 | 0.31 | 0.62 | 0.61 | 0.57 | 0.51 | 0.35 | 0.69 | 0.63 | 0.54 |

*Table 7.* Ablation on the criterion used to select the vision layer $l$ in the dual embedding.

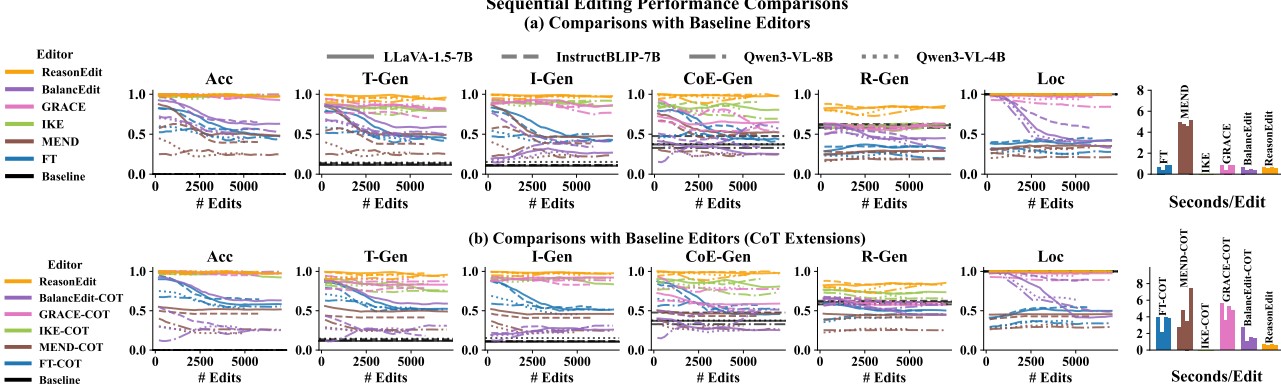

*Figure 12.* Sequential editing performance and efficiency across editors. ReasonEdit achieves the best sample generalities, rationale-informed generalities, as well as high reliability and locality. Trajectories are smoothed using moving average with a 5-step window.

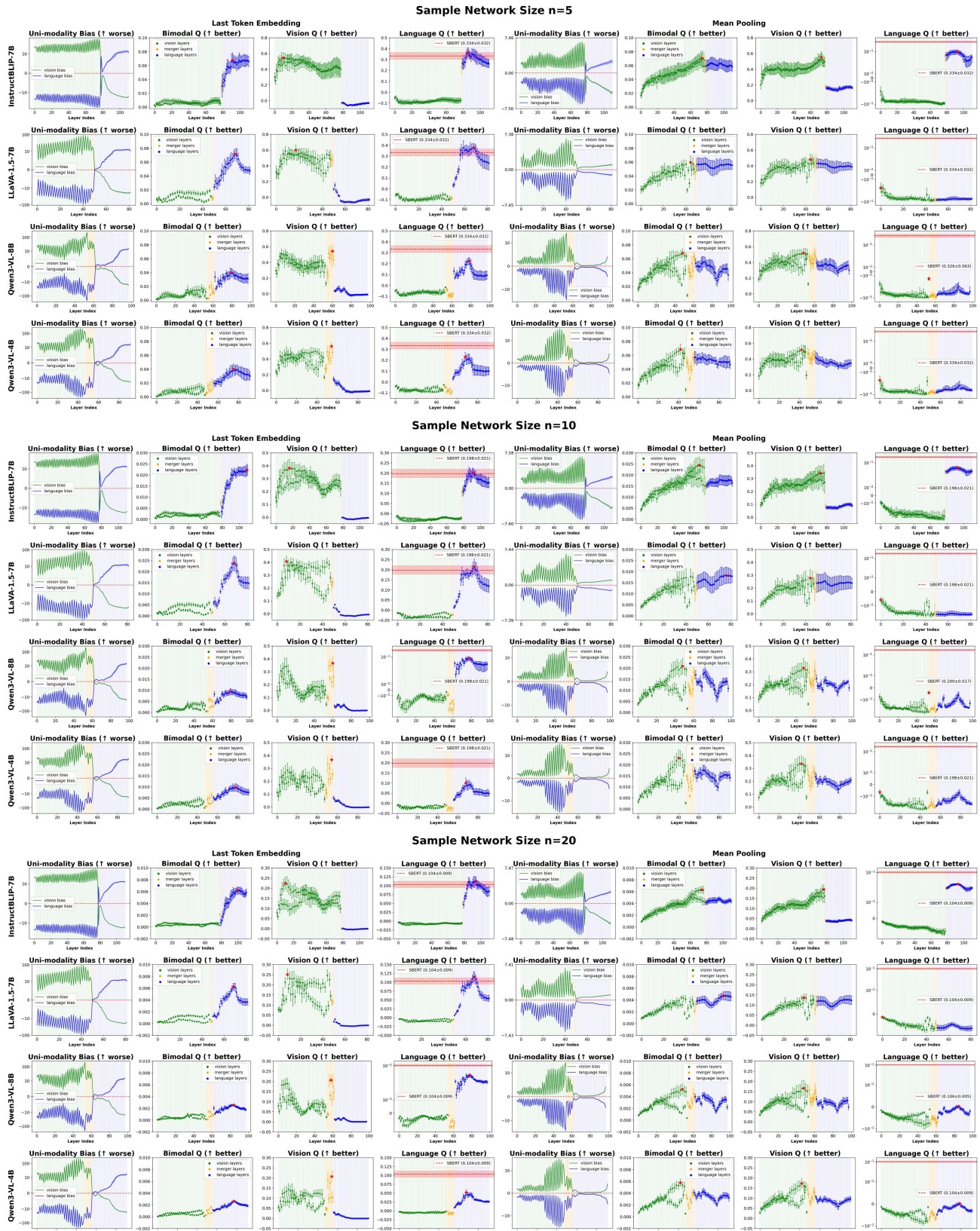

*Figure 13.* The number of image–text pairs $n$ used to construct sample networks changes the scale of the metrics for topology-aware multi-modal embedding evaluation, but preserves their ordering and relative differences across embeddings.

