# OpenReview forum: "ReasonEdit: Editing Vision--Language Models using Human Reasoning"
_ICML.cc/2026/Conference — ICML 2026 regular_

### Official Review · Reviewer_b2vx · 2026-03-08

**Soundness:** 3
**Presentation:** 1
**Significance:** 1
**Originality:** 2
**Overall Recommendation:** 2
**Confidence:** 4

**Summary:**

This paper proposes a method for VLM model editing based on retrieval-based methods. Rather than a human providing a single label or factual association as the correction, the human provides a reasoning chain (a chain-of-thought) of factual statements, reasoning about the answer. As a result, rather than retrieving the correct answer, it retrieves the underlying reasoning chains that can compose to result in a correct answer. To construct the codebook, the user asks the VLM a question about the image, and the VLM gives a wrong answer. The user then provides the correct answer + a chain of reasoning for that answer. The user-provided chain is broken down into N_s individual parts, each paired with visual grounded evidence in the image (image parts). The codebook contains keys and associated values. The keys are multimodal embeddings of the image-text pairs. The pairs can either be the multimodal embeddings of (full image, question), denoted as answer entries, with corresponding values representing a templated text of (question, direct answer), or it can be (part of image, part of reasoning chain s_j) with associated value being s_j, and these entries are denoted as reasoning entries. If there exists a similar edit , the keys are merged by averaging and the values (text) are concatenated. At inference time, given a query, the method perform a nearest neighbor search between the query input question and all keys in the codebook and if the minimum distance is bigger than all existing entries in the codebook, no retrieval is performed, and otherwise, the method retrieves the K values (which are K sentences where each can either be s_j or the templated direct answer), concatenates all K into a single sentence, and uses that as context to the prompt of the VLM.  To make retrieval work well, they introduce a topology-balanced multimodal embedding that tries to avoid being biased toward only the image or text modality, preventing one modality from dominating in the retrieval.

**Compliance With Llm Reviewing Policy:**

Affirmed.

**Key Questions For Authors:**

I apologize for raising a lot of weaknesses, but the paper in its current form is not ready. I am mostly worried about W1, W2, W4, W6 and W7. For this reason, my decision will sadly be negative.

**Limitations:**

I did not see any section on Limitations.

**Strengths And Weaknesses:**

Strengths:

- Nice idea of using a compositional codebook; parts of reasoning chains which removes the limitation of a single query-corrected answer association, and now opens up the possibility to compose multiple chains to form an answer. Therefore, a single edit can now utilize multiple related chains (possibly from different editing samples) and compose them to edit an answer, helping generalization.
- The motivation is strong, specifically applying the works from text-only LLMs (e.g., IKE) fail. Specifically, the authors mention that in VLMs, the 1) edited information may refer to image patches which is more challenging, 2) weight-update methods are hard to work, and 3) one modality dominates which is a big failure. All these problems are well-tackled in the paper.

Weaknesses:
- [W1] At inference, the query itself is the whole input pair: the full image + the question, therefore the retrieval can easily overfit to the answer entries rather than the reasoning entries, since the answer entries are constructed via the full image (as in the case in inference time). This renders the method not that different from previous works, deviating away from the whole purpose of the work.
- [W2] The readability is very bad. The Figures are hard to read, too much information. Figures 4 and 5 cannot be followed. This is a serious issue because readability is as important as the quality of the work. I am giving a score of 1 for presentation.
- [W3] The idea is very similar to [R1], just applied in the task of editing.
- [W4] I really think this work is another way of doing Visual Chain of Thought Reasoning. Some representative papers are [R2-R3]. Both works require human annotation of multimodal chain of thoughts, but one trains/finetunes a VLM, and the other (model editing) stores embeddings - which is a way of training, since the stored embeddings are a kind of few-shot training examples). In essence there is a big overlap in these two fields. For this reason, I am giving a score of 1 for significance. To prove this is a good strategy (either giving better performance or giving comparable performance with better training and testing efficieny), the authors should report the overall downstream VQA accuracy on the full A-OKVQA or FVQA benchmark after editing. The Acc metric reported is the success on edited/error cases, not the model’s final full-dataset A-OKVQA/FVQA accuracy after applying edits. To show that editing is actually helpful in practice, the authors should show the final  benchmark accuracy improved.
- [W5] The method requires an external sentence embedding model (paraphrase-mpnet-base-v2). This just adds more complexity.
- [W6] If the user does not manually provide visual evidence, it is very computationally expensive to find such regions as shown in Figure 10. Many regions have to be tested, and I do not think this is a clever solution. Why does it have to be a "brute-force” way of "try everything"? Can’t you use attention maps for example to find a related region or at least to restrict the search space? Ive played a lot with [R5] and found it to provide very good attention maps localizing a query for VLMs. The authors can investigate other ways, this is just one way from the top of my head.
- [W7] The datasets used (e.g., FVQA and A-OKVQA) are typically not reasoning benchmarks. I would expect reasoning benchmarks to be benchmarks with logical or mathematical reasoning, such as LogicVista, MathVista, and others.


[R1] DCoT: Dual Chain-of-Thought Prompting for Large Multimodal Models\
[R2] VoCoT: Unleashing Visually Grounded Multi-Step Reasoning in Large Multi-Modal Models\
[R3] Pixel Reasoner: Incentivizing Pixel-Space Reasoning with Curiosity-Driven Reinforcement Learning\
[R4] Visual CoT: Advancing Multi-Modal Language Models with a Comprehensive Dataset and Benchmark for Chain-of-Thought Reasoning\
[R5] Your Large Vision-Language Model Only Needs A Few Attention Heads For Visual Grounding

---

> ### Author Rebuttal · Authors · 2026-03-30
>
> Thank you for the feedback. We **respectfully disagree** with many of the reviewer’s points. While we have tried to improve our paper based on your concerns, we find several points unclear, and several pertain to settings unrelated to model editing.
> - The reviewer states our work is “not that different from previous model editors (W1)” while stating our work is “really another way to train CoT models (W4)”. These points either seem inconsistent with each other or **equate** model training with post-training model editing. We therefore first show that the two fields are **fundamentally different**:
>
> ||Inference Time|One-shot|Targeted Error Correction|Generality|Locality|Reasoning-enabled New Generality|
> |-|-|-|-|-|-|-|
> |Pretrain/Finetune(FT) Foundation Models|X|X Large-scale data required|X|X|X|X FT-based editors perform poorly (Table 1)|
> |Model Editing|√|√|√|√ Generalize each edit to unseen samples|√ Prevent model degradation; Preserve unrelated behaviors|√ Introduced by our work|
>
> W1. The query is full image+question, retrieval overfit answer.
> - Using answers during editing and querying with full image + question is **standard** in model editing to verify successful error correction (Acc~100%). The core challenge is the **generality–locality tradeoff**, and our work further enables new forms of reasoning-enhanced generality. To better clarify, we have added a new ablation study to remove answers during editing (please see results in reviewer r8Sp W3). We find ReasonEdit approaches the ceiling performance where VLMs infer answers from ground-truth human reasoning.
>
> W2. The readability is very bad. Presentation score 1.
> - **We are very surprised to see this.** The reviewer’s summary describes nearly every detailed step in the method pipeline accurately at high granularity. While more specific feedback would be helpful, we have tried to improve clarity by reducing line overlap in Fig. 4 and increasing font size in Fig. 5.
>
> W3. The idea is very similar to DCoT, just applied in the task of editing.
> - **Respectfully, the similarity is minimal.** We find that this feedback can be given to any work using context retrieval (e.g., IKE, RAG, VisRAG) or visual grounding (e.g., MSCKE, RegionCLIP, GLIP). To address your feedback, DCoT specifically retrieves demo QAs from 8 problem categories as context to guide generation, but we retrieve human reasoning facts + visual evidence to correct model errors. **Inputs, methods, and objectives all differ largely.**
>
> W4.1 This work is just doing Visual CoT Reasoning (e.g., VoCoT, Pixel Reasoner).
> - We kindly ask the reviewer to be more specific on “doing Visual CoT Reasoning.” These terms can describe a large variety of problems / methods. To address your feedback, we summarize the key differences below. VoCoT and Pixel Reasoner train foundation reasoning models and are **unrelated to model editing.**
> ||Setting|One-shot|User-interactive|Correct errors|Sequential editing|Generality-locality tradeoff|
> |-|-|-|-|-|-|-|
> |VoCoT|Training|X (80K samples, multi-stage instruction tuning)|X|X|X|X|
> |Pixel Reasoner|Training|X (15K queries, instruction tuning + reinforcement learning)|X|X|X|X|
> |Model Editors (BalancEdit, IKE, GRACE, MEND, ReasonEdit)|Post-training|√|√|√|√|√|
>
> W4.2  In essence there is a big overlap in these two fields. For this reason, significance score 1.
> - “Model editing” and “train reasoning foundation models” are clearly distinct fields with unique objectives. Respectfully, we further find it unclear why overlap between two **fields** suggests low significance of our work.
>
> W4.3 To prove strategy good, report overall accuracy on A-OKVQA or FVQA.
> - The overall accuracy is already in our results as the combination of Acc (edit set) and Loc (remaining set). For ReasonEdit, ~100% Acc and 100% Loc result in ~100% overall accuracy. We have added the results below in Sec 4.3.
> ||LLaVA|InstructBLIP|Qwen4B|Qwen8B|
> |-|-|-|-|-|
> |FVQA|1.000|0.995|0.998|0.994|
> |A-OKVQA|0.996|0.997|0.991|0.992|
>
> W5 Sentence encoder just adds complexity.
> - The sentence encoder provides pure-text similarity at much lower dimension (reduced storage) than language layers. It is widely used in prior editors (BalancEdit, IKE) with negligible compute and memory overhead.
>
> W6 Visual evidence is very computationally expensive and not clever, why not attention mapping or others.
> - We would like to clarify that acquiring visual evidence is **not computationally expensive**. In Sec. 4.4 (line378–380), “ReasonEdit takes the same amount of time as the non-CoT editors, but is significantly faster than their CoT extensions.” Attention mapping may be an alternative but require extra hyperparameters (e.g., model-specific layer selection, heads aggregation).
>
> W7 FVQA and A-OKVQA are not reasoning benchmarks. I would use logic or math.
> - Both FVQA and A-OKVQA provide human-written, natural image-grounded reasoning on which our method depends. They are **sufficient for comparing VLM editors on a common testbed**.

---

> > ### Author Rebuttal · Reviewer_b2vx · 2026-04-01
> >
> > I thank the authors for their rebuttal. However, I did not appreciate the tone of the rebuttal, which was defensive rather than actionable. This is not what the rebuttal is meant for. I've put an effort of 2 days to read and understand this paper as well as other similar works in model editing, and I wrote a constructive review. However, the authors did not address the concerns and rather took a defensive approach, disagreeing on many points I raised and refusing to conduct any other experiments, replying me with statements like "our existing experiments are sufficient for comparing VLM editors on a common testbed", or "the similarity is minimal with related works", while I would have expected the authors to conduct experiments. This paper is about reasoning, and should be evaluated on reasoning datasets, and especially in the cases where human interference is required, but FVQA and A-OKVQA are not the case for this. DCoT is also retrieving reasoning templates, just like the authors do (but the authors mention that it only retrieves demo QAs, please read the paper well).  Other concerns I raised like the usage of the sentence encoder are addressed by a defensive approach of: "negligible compute and memory overhead." rather than actually showing the time/memory/compute comparison, or at least the authors could have addressed this by using the VLM itself to encode. But no experiment was conducted. W6 is also not addressed but rather defended by: "not computationally expensive", without showing any compute times on this specific part, which I expected. I also did not appreciate the authors pretending to not understand my concern on readability, saying that they are "surprised" that i raised this. It is clear from my comments that the readability of the Figures is very bad, with too much information, making it very hard to get conclusions out of the Figures . For example, Figure 4 has 20 curves, and is 1/8 of the page width.
> >
> > In general, the rebuttal did not address my concerns and was defensive rather than actionable. For this reason, I maintain my score as a reject. I will also not engage with the authors anymore because I did not appreciate the tone of the rebuttal.

---

### Official Review · Reviewer_zARE · 2026-03-12

**Soundness:** 3
**Presentation:** 3
**Significance:** 3
**Originality:** 3
**Overall Recommendation:** 5
**Confidence:** 3

**Summary:**

This paper studies model editing for vision-language models (VLMs), focusing on reasoning-heavy VQA scenarios. The authors propose ReasonEdit, a framework that allows users to provide step-by-step reasoning during editing. The method decomposes edits into reasoning-level entries aligned with visual evidence patches and stores them in an external codebook. During inference, relevant reasoning entries are retrieved using a topology-balanced dual embedding and appended to the prompt to guide model predictions without updating model parameters. Experiments on A-OKVQA and FVQA across four VLMs show that ReasonEdit significantly outperforms several existing editing approaches in editing success rate, locality, and two newly proposed reasoning generalization metrics.

**Compliance With Llm Reviewing Policy:**

Affirmed.

**Final Justification:**

My concerns have been adequately addressed.

**Key Questions For Authors:**

See the Weaknesses.

**Limitations:**

yes

**Strengths And Weaknesses:**

### Strengths

1. The paper extends model editing to a reasoning-aware setting, where users can provide explicit reasoning steps during editing. Compared to traditional editing methods that only provide answers, this setting better reflects realistic interaction scenarios and opens up new directions for incorporating human reasoning signals into model editing.

2. The evaluation covers multiple VLM architectures and two reasoning-based VQA benchmarks. The authors analyze editing reliability, locality, sequential editing, and efficiency. The proposed method outperforms baselines across most settings. The paper also introduces two new metrics (R-Gen and CoE-Gen) intended to evaluate reasoning-level generalization.

### Weaknesses

1. This paper introduces the R-Gen and CoE-Gen metrics to measure the generalization ability of reasoning facts. However, ReasonEdit explicitly stores and retrieves fact-level reasoning entries, which may naturally benefit such evaluations. It remains unclear whether these metrics can fairly compare different types of editing methods. Additional validation on more standard editing benchmarks or alternative evaluation settings would strengthen the conclusions.

2. ReasonEdit assumes that users can provide relatively accurate and structured reasoning statements. However, human-provided reasoning can be noisy, incomplete, or partially incorrect. Since ReasonEdit's memory entries directly depend on these reasoning statements, the method may be sensitive to the quality of the reasoning. This paper does not investigate the robustness of the method under supervision with noisy or imperfect reasoning.

---

> ### Author Rebuttal · Authors · 2026-03-30
>
> Thank you for the constructive suggestions, and for recognizing the practical value of our problem setting, the thoroughness of our evaluation, and the clear presentation of our work. Your valuable insight has strengthened our work from new perspectives!
>
> - **W1. Clarify R-Gen and CoE-Gen and add alternative evaluation settings.**
>
> We first clarify that fact-level reasoning entries are provided **consistently across all editors in their CoT-enabled versions**, not only ReasonEdit. For weight-updated editors (FT, MEND, BalancEdit), they are provided during training. For retrieval-based editors (IKE, GRACE), the same entries are stored in the codebook. We intend for reasoning facts to be learned or stored during editing and retrieved as context at inference. Under the fair comparison, R-Gen evaluates whether an edited VLM can accurately predict nuanced reasoning facts at inference, while CoE-Gen evaluates whether an edited VLM can accurately use relevant reasoning facts as context in unseen scenarios. All generality evaluations use unseen new samples.
>
> Thanks to the suggestions by you and reviewer r8Sp, we have added two new evaluation settings: (1) held-out error evaluation for reasoning-enabled distribution drift; and (2) ablation study on answer-only and reasoning-only entries in the codebook. To use the remaining space to address your other question, please see our response to reviewer r8Sp W1 & W3 for the full results. From these experiments, we find that ReasonEdit maintains good performance under these alternative evaluation settings.
>
> - **W2. Investigate the robustness of ReasonEdit under noisy or imperfect reasoning.**
>
> Thanks for the great insight! We assume human reasoning is reliable ground truth in our paper, but this is a valuable additional experiment that improves current work and highlights an important direction for future studies.
>
> Therefore, we have added 2 new experiments to evaluate robustness under noisy reasoning as a new section in the paper. During editing, for each edit, we (1) inject or (2) replace 10%, 30%, and 50% of its reasoning statements with noise (irrelevant random sentences). The results using the combined datasets are shown below.
>
> - **(1) Inject noise to reasoning.**
>
> |LLaVA|Acc|I-Gen|T-Gen|R-Gen|CoE-Gen|Loc|
> |-|-|-|-|-|-|-|
> |no noise|1.00|0.97|0.97|0.86|0.98|1.00|
> |10%|0.99|0.98|0.97|0.83|0.99|1.00|
> |30%|0.99|0.98|0.97|0.83|0.98|1.00|
> |50%|0.99|0.97|0.98|0.82|0.99|1.00|
>
> |InstructBLIP|Acc|I-Gen|T-Gen|R-Gen|CoE-Gen|Loc|
> |-|-|-|-|-|-|-|
> |no noise|0.98|0.95|0.96|0.87|0.96|1.00|
> |10%|0.99|0.95|0.96|0.84|0.97|1.00|
> |30%|0.98|0.95|0.96|0.83|0.98|1.00|
> |50%|0.99|0.95|0.95|0.83|0.97|1.00|
>
> |Qwen4B|Acc|I-Gen|T-Gen|R-Gen|CoE-Gen|Loc|
> |-|-|-|-|-|-|-|
> |no noise|0.99|0.91|0.91|0.80|0.96|1.00|
> |10%|0.95|0.93|0.91|0.81|0.97|1.00|
> |30%|0.96|0.93|0.92|0.80|0.97|1.00|
> |50%|0.96|0.93|0.92|0.81|0.97|1.00|
>
> |Qwen8B|Acc|I-Gen|T-Gen|R-Gen|CoE-Gen|Loc|
> |-|-|-|-|-|-|-|
> |no noise|0.98|0.94|0.92|0.80|0.95|1.00|
> |10%|0.98|0.94|0.91|0.78|0.98|1.00|
> |30%|0.99|0.94|0.92|0.78|0.97|1.00|
> |50%|0.99|0.94|0.92|0.78|0.97|1.00|
>
> - **(2) Replace reasoning with noise.**
>
> |LLaVA|Acc|I-Gen|T-Gen|R-Gen|CoE-Gen|Loc|
> |-|-|-|-|-|-|-|
> |no noise|1.00|0.97|0.97|0.86|0.98|1.00|
> |10%|0.99|0.97|0.97|0.83|0.98|1.00|
> |30%|0.99|0.98|0.97|0.81|0.94|1.00|
> |50%|1.00|0.98|0.97|0.77|0.93|1.00|
>
> |InstructBLIP|Acc|I-Gen|T-Gen|R-Gen|CoE-Gen|Loc|
> |-|-|-|-|-|-|-|
> |no noise|0.98|0.96|0.95|0.87|0.96|1.00|
> |10%|0.97|0.95|0.95|0.83|0.97|1.00|
> |30%|0.98|0.95|0.95|0.80|0.96|1.00|
> |50%|0.98|0.95|0.95|0.77|0.95|1.00|
>
> |Qwen4B|Acc|I-Gen|T-Gen|R-Gen|CoE-Gen|Loc|
> |-|-|-|-|-|-|-|
> |no noise|0.99|0.91|0.91|0.80|0.96|1.00|
> |10%|0.96|0.93|0.91|0.79|0.95|1.00|
> |30%|0.96|0.92|0.91|0.77|0.93|1.00|
> |50%|0.96|0.93|0.90|0.76|0.92|1.00|
>
> |Qwen8B|Acc|I-Gen|T-Gen|R-Gen|CoE-Gen|Loc|
> |-|-|-|-|-|-|-|
> |no noise|0.98|0.94|0.89|0.80|0.95|1.00|
> |10%|0.98|0.94|0.91|0.77|0.94|1.00|
> |30%|0.96|0.94|0.89|0.76|0.92|1.00|
> |50%|0.98|0.93|0.90|0.74|0.90|1.00|
>
> We find that ReasonEdit is very robust to the noise injected into reasoning, as all performances are close to those under noise-free reasoning. When real reasoning facts are replaced with noise, R-Gen and CoE-Gen degrade as expected, since the codebook is missing relevant facts. This highlights the crucial role of reasoning facts in reasoning-enabled generalization for model editing.
>
> Based on your feedback, we have also added the following to a limitation section: “ReasonEdit assumes users provide accurate and structured reasoning. However, human reasoning can be noisy, incomplete, or incorrect. The current paper provides an initial evaluation of robustness under noisy reasoning, but a more comprehensive study remains an important direction for future studies.”

---

> > ### Author Rebuttal · Reviewer_zARE · 2026-04-03
> >
> > The authors have fully addressed my concerns, and I will raise my score.

---

> > > ### Author Response · Authors · 2026-04-04
> > >
> > > Thank you for acknowledging our rebuttal and for raising your score!  We appreciate your time and feedback.

---

### Official Review · Reviewer_r8Sp · 2026-03-13

**Soundness:** 2
**Presentation:** 3
**Significance:** 3
**Originality:** 3
**Overall Recommendation:** 4
**Confidence:** 3

**Summary:**

This paper introduces ReasonEdit, a model editing method for VLMs that allow user to provide human reasoning (a few factual statements) when correcting/editing model. The proposed method includes three key components: 1) a codebook that stores key-value (K-V) pairs, where the key embeds the textual content and relative visual content, and the value records the corresponding factual text (reasoning statement or question-answer pair); 2) a codebook expansion procedure that iteratively insert and merge new K-V pairs from incoming new edit; 3) a network topology-aware method to calculate multimodal embeddings for retrieving codebook keys. At inference time, the method retrieve relevant facts in the codebook via kNN and use them as multimodal prompt prefixes. Experiments show that the proposed editing framework performs good on FVQA and A-OKVQA across four evaluated VLMs.

**Compliance With Llm Reviewing Policy:**

Affirmed.

**Final Justification:**

In the initial reviews, I raised several concerns on the generalizablity of the proposed method, especially on soundness (hyperparameter generalizablity and whether the method have memorized most facts in the codebook). In the author's rebuttal, they addressed most of my concerns. Therefore, I lean to weak accept.

**Key Questions For Authors:**

1. Could you provide results with a held-out error split? E.g., edit a fraction of errors and evaluate R-Gen/CoE-Gen on the remaining that share reasoning patterns with edited ones but were never edited? This would be the most direct way to validate the "reasoning improves generalization" claim.

2. Have you verified that the selected l^\* and w^\* transfer across datasets? E.g., select on ImageNet but test on COCO-based question subset, and vice versa? If the optimal hyperparameters differ significantly, it would suggest the topology-aware criterion is dataset-specific rather than principled.

3. Could you show the empirical results of using different the codebook entry types? Additionally, could you compare $Q_{bi}$ against $Q_{vis}$ or $Q_{lang}$ as the layer selection criterion?

**Limitations:**

It would be beneficial to further discuss the limitations. E.g., the dependence on the quality and availability of human reasoning annotations; and the evaluation gap between the current "error correction" setting and realistic model editing scenarios like  adapting distribution shift or knowledge updates over time.

**Strengths And Weaknesses:**

### Strengths

1. The problem formulation is well-motivated and practical. Allowing users to explain why an answer should change and provide corresponding rationales, rather than just providing the corrected label, is a natural paradigm for model editing.

2. The topology-aware embedding selection is a creative idea. Using Newman modularity from network science to diagnose "visual bias" vs. "language bias" in different VLM layers is interesting. Empirically, experiments shown in Figure 6 do demonstrate a trade-off of believing more on visual or language embeddings for related fact retrieval.

3. The paper is generally well-written and clearly structured. The method figures are helpful to understand the overall method.

### Weaknesses

1. The evaluation protocol seems weaker than prior retrieval-based editing work. The paper's experiment setting edits **all** incorrectly-answered samples and evaluates on these same samples and their derivatives, with no held-out error split. For a retrieval-based method, near-perfect Reliability (0.99-1.00) is expected and uninformative, since the method stored all the answers in the codebook and could retrieve them back at inference time. Compared to GRACE, which also uses a codebook-based approach, the experimental design here seems less reasonable. GRACE evaluates under distribution shift (e.g., different answer label distribution). In contrast, ReasonEdit operates entirely within the same VQA dataset, where the edit set and evaluation set are derived from the same dataset with no distribution shift. This setting is more like "correcting errors of VLM" rather than "editing updated facts in VLM", and making it difficult to assess whether the method would work in realistic scenarios.

2. Hyperparameter selection seems to be using test-distribution data. The topology-aware embedding selection (l^\*, w^\*) samples image-text pairs from COCO and ImageNet (Sec 4.2, Appendix D.4), which are exactly the image sources of FVQA (COCO-2014 + ImageNet) and A-OKVQA (COCO-2017). While this doesn't directly use edit labels, the embedding space and hyperparameters are already optimized for the test task's input distribution. This raises a question on the validity and generalizability of selected hyperparameters for real **unseen image domains**.

3. Lack of ablations on key design choices. Several important design decisions are not empirically ablated:

   - *Topology criterion selection*: The paper uses $Q_{bi}$ to select the vision layer $l^*$, but does not compare against using $Q_{vis}$ or $Q_{lang}$ as the selection criterion. It is unclear whether $Q_{bi}$ is genuinely better, or simply happens to work on these two datasets.

   - *Reasoning entries vs. answer entries*: The codebook stores both answer entries (1 per edit) and reasoning entries (multiple per edit, one per fact-patch pair). The paper argues that human reasoning is the key driver of improved generalization, but does not ablate the contribution of each entry type. What happens if the codebook only stores answer entries (no reasoning) or only reasoning entries (no direct answer)?

---

> ### Author Rebuttal · Authors · 2026-03-30
>
> Thank you for the **very constructive** feedback and for appreciating our creative topology-aware multimodal embedding strategy (and our clear presentation and motivation)! Based on your feedback, we have added several new experiments to the paper, which improves it in many ways.
>
> - **W1. Held-out error evaluation (a novel distribution shift experiment)**
>
> Thank you for the idea! We first clarify that the generality and locality evaluation datasets consist of **unseen new samples**, instead of the edit set. Especially, the unseen samples used to evaluate R-Gen and CoE-Gen are only partially related to the edit set through shared reasoning facts. We have better clarified this in Sec. 4.1.
>
> Your suggestion introduces **a novel reasoning-enabled distribution shift** to the standard model editing evaluation, which can make ReasonEdit an even stronger contribution. Similar to GRACE, we added the new experiment: for each VLM, we embed each reasoning fact using a sentence encoder and group edits that share at least one fact with similarity > 0.95. Stratified by groups, we split edits 50/50 into an edit set and a held-out set for evaluation.
> |edits(groups)|LLaVA|InstructBLIP|Qwen4B|Qwen8B|
> |-|-|-|-|-|
> |FVQA|482(120)|446(117)|386(112)|508(131)|
> |AOKVQA|2021(498)|1486(417)|1216(351)|2022(502)|
> |**Held-out Acc**|||||
> |FVQA|0.85|0.86|0.83|0.80|
> |AOKVQA|0.79|0.78|0.75|0.73|
> |**Held-out R-Gen**|||||
> |FVQA|0.81|0.83|0.79|0.76|
> |AOKVQA|0.76|0.72|0.69|0.68|
> |**Held-out CoE-Gen**|||||
> |FVQA|0.84|0.82|0.82|0.80|
> |AOKVQA|0.76|0.75|0.74|0.71|
>
> The held-out Acc (baseline 0%) is high on both datasets. For held-out R-Gen and CoE-Gen, ReasonEdit remains substantially higher than the unedited baseline on both FVQA and A-OKVQA. This shows ReasonEdit also enables new generalization in this stronger setting.
>
> - **W2. Verify the topology-aware embedding strategy is data-agnostic.**
>
> This is another great idea and further strengthens our topology-balanced embedding method. We compare hyperparameter selections across 3 image sources.
> |$l$ index|LLaVA|InstructBLIP|Qwen4B|Qwen8B|
> |-|-|-|-|-|
> |COCO|19|38|21|24|
> |ImageNet|19|38|20|24|
> |Flickr30k (new)|19|38|20|23|
> |**$w$**|||||
> |COCO |7|8|41|18|
> |ImageNet|8|8|40|18|
> |Flickr30k (new)|7|8|40|19|
>
> The high consistency of these hyperparameters (close layer indices and balancing weights) indicates our topology-aware selection is empirically **data-agnostic**.
>
> W3. Two additional ablation studies.
>
> - ***reasoning entries* vs. *answer entries*.** To isolate the effects of answer and reasoning on editing performance, we added an ablation that keeps only answer or reasoning entries in the codebook, using combined datasets. As shown below, (1) using answer entries, ReasonEdit’s performance is close to the upper bound where the ground truth (GT) answer is directly provided in the prompt. The lower R-Gen and CoE-Gen is expected since no reasoning is provided. (2) When using reasoning entries, we observe that ReasonEdit’s performance approaches the ceiling performance (upper bound) where VLMs infer the answer from ground-truth human reasoning, indicating reliable retrieval of reasoning.
> |LLaVA|Acc|I-Gen|T-Gen|R-Gen|CoE-Gen|
> |-|-|-|-|-|-|
> |prompt GT answer(ceiling)|1.0|0.95|0.99|0.67|0.76|
> |edit w/ answer entries only|0.98|0.89|0.96|0.65|0.72|
> |prompt GT reason(ceiling)|0.85|0.84|0.90|0.92|0.89|
> |edit w/ reason entries only|0.81|0.81|0.80|0.87|0.86|
> |**InstructBLIP**||||||
> |GT answer ceiling|1.0|0.91|0.95|0.64|0.79|
> |answer entries|0.99|0.85|0.91|0.62|0.75|
> |GT reason ceiling|0.83|0.86|0.78|0.94|0.87|
> |reason entries|0.80|0.74|0.75|0.87|0.83|
> |**Qwen4B**||||||
> |GT answer ceiling|1.0|0.95|0.90|0.61|0.71|
> |answer entries|0.98|0.91|0.87|0.59|0.69|
> |GT reason ceiling|0.79|0.81|0.77|0.91|0.82|
> |reason entries|0.76|0.77|0.75|0.83|0.79|
> |**Qwen8B**||||||
> |GT answer ceiling|1.0|0.92|0.90|0.61|0.70|
> |answer entries|0.98|0.88|0.87|0.61|0.67|
> |GT reason ceiling|0.77|0.80|0.75|0.89|0.80|
> |reason entries|0.76|0.76|0.73|0.84|0.78|
>
> - **Select l by $Q_{vis}$ vs $Q_{lang}$.** We first clarify that $Q_{bi}$ aims to find $l$ with more balanced topology among the already vision-biased layers. We have added an ablation study as below. We find that $Q_{vis}$ yields similar performances to $Q_{bi}$ with lower I-Gen, while $Q_{lang}$ results in substantially worse performance, due to the dual embedding being now completely text-biased.
> |LLaVA|Acc|T-Gen|I-Gen|R-Gen|CoE-Gen|
> |-|-|-|-|-|-|
> |$Q_{bi}$|1.00|0.97|0.97|0.86|0.98|
> |$Q_{vis}$|0.99|0.96|0.93|0.84|0.95|
> |$Q_{lang}$|0.76|0.78|0.80|0.72|0.74|
> |InstructBLIP||||||
> |$Q_{bi}$|0.98|0.96|0.95|0.87|0.96|
> |$Q_{vis}$|0.96|0.95|0.91|0.85|0.95|
> |$Q_{lang}$|0.73|0.74|0.79|0.70|0.79|
> |Qwen4B||||||
> |$Q_{bi}$|0.99|0.91|0.91|0.83|0.96|
> |$Q_{vis}$|0.98|0.90|0.89|0.78|0.91|
> |$Q_{lang}$|0.51|0.35|0.69|0.63|0.54|
> |Qwen8B||||||
> |$Q_{bi}$|0.98|0.89|0.94|0.84|0.95|
> |$Q_{vis}$|0.97|0.88|0.91|0.78|0.94|
> |$Q_{lang}$|0.57|0.31|0.62|0.61|0.57|
>
> Thank you! We have added a detailed limitation section.

---

> > ### Author Rebuttal · Reviewer_r8Sp · 2026-04-04
> >
> > Thanks for detailed response to my concerns. All of these have resolved my initial concerns. I would increase my score accordingly.

---

> > > ### Author Response · Authors · 2026-04-04
> > >
> > > Thank you for reading our rebuttal and for raising your score! We appreciate your time and feedback.

---

### Official Review · Reviewer_Y3kq · 2026-03-13

**Soundness:** 3
**Presentation:** 3
**Significance:** 2
**Originality:** 3
**Overall Recommendation:** 4
**Confidence:** 3

**Summary:**

This paper introduces ReasonEdit, the first reasoning-enhanced model editing method designed specifically for Vision–Language Models (VLMs). The authors address three key challenges in VLM editing: aligning fine-grained visual details with textual reasoning, avoiding catastrophic forgetting in weight-updating methods, and the ad-hoc selection of layers for editing. ReasonEdit proposes a retrieval-based approach that stores image-text queries paired with human reasoning statements in a codebook. During inference, relevant reasoning is retrieved and prepended as context. A novel topology-balanced dual embedding method is introduced to improve retrieval by balancing vision and language biases, using modularity-based graph analysis to select embeddings. Experiments on four VLMs and two VQA datasets show that ReasonEdit outperforms prior methods in generalization, locality, and efficiency, especially in sequential editing and reasoning-based generalization tasks.

**Compliance With Llm Reviewing Policy:**

Affirmed.

**Key Questions For Authors:**

1. How robust is the automatic patchification process when no user-provided patches are available? Could you provide a quantitative evaluation of its accuracy or impact on retrieval performance?
2. The method relies heavily on human reasoning. How would ReasonEdit perform in scenarios where such reasoning is unavailable, noisy, or incomplete? Have you tested any fallback strategies?
3. In sequential editing, what happens when the codebook grows very large? Are there mechanisms for pruning or forgetting outdated edits?

**Limitations:**

yes

**Strengths And Weaknesses:**

Strengths:
The paper is technically sound. The authors provide a clear problem formulation and propose a well-motivated retrieval-based solution. The experimental design is thorough, covering four VLMs, two datasets, and multiple evaluation metrics (reliability, locality, text/image/rationale/CoE generality). The ablation studies are comprehensive, especially the analysis of embedding biases and key merging.

Weaknesses:
- The automatic patchification process for visual evidence (when not user-provided) is described but not rigorously evaluated.

- The reliance on GPT-4o and diffusion models for generating evaluation data (e.g., R-Gen, CoE-Gen) may introduce noise; the authors perform a quality check but do not quantify its impact.

- The paper does not discuss potential failure cases or limitations of the retrieval-based approach (e.g., when no relevant key exists).

typo:
- The paper contains repetitive sentences (e.g., duplicated contributions in Section 1).
- Some figures (e.g., Fig. 4, Fig. 5) are referenced but not clearly explained in the text.

---

> ### Author Rebuttal · Authors · 2026-03-30
>
> Thank you for the constructive suggestions and for recognizing our paper as technically sound, clearly formulated, and well motivated. We also appreciate your recognition of our first reasoning-enhanced VLM editing method and our novel topology-balanced multimodal embedding design. Your suggestions have improved our work in many ways.
>
> - **W1 (Q1). Directly evaluate the automatic patchification process.**
>
> This is a great suggestion. A direct evaluation of automatic patchification can further strengthen our current experiments. Based on your feedback, we have added a new experiment as follows. We compute CLIP similarity between each automatically extracted patch and its associated reasoning text using the edit set from each VLM. We compare it with the baseline CLIP similarity estimated between the raw images and their ground-truth captions. Results are shown below.
>
> | CLIP similarity ↑ | LLaVA | InstructBLIP | Qwen4B | Qwen8B |
> |---|---|---|---|---|
> | Raw Images (baseline) | 0.269 (0.041) | 0.260 (0.042) | 0.273 (0.038) | 0.261 (0.043) |
> | Auto Patches | 0.277 (0.036) | 0.267 (0.036) | 0.274 (0.037) | 0.273 (0.036) |
>
> We find that compared to the baseline, our automatic patchification achieves a similar level of CLIP similarity. This indicates the patchification process preserves semantic alignment with the associated reasoning text.  We have added this evaluation to Section 3.2 and Appendix C.
>
> - **W2. Add quantitative impact of the quality check on the generated data.**
>
> We would like to first clarify that, in line 926 paragraph “Image Generation Quality Evaluation”, we provided details on how we conduct a three-round evaluation on each of the generated images, and reported the proportion of the affected samples that failed the check. Based on your feedback, we have made this more concrete by adding the following details: “This quality check removes 389 generated images for A-OKVQA and 74 for FVQA after up to three regeneration attempts.”
>
> - **Q2.  Assess robustness of ReasonEdit under noisy or imperfect reasoning.**
>
> Thank you for this **valuable insight**! Based on your feedback, we have added two new experiments: during editing, for each edit, we (1) inject or (2) replace 10%, 30%, and 50% of its reasoning statements with noise (irrelevant random sentences). Since reviewer zARE raised a similar question, we refer to our response to reviewer zARE W2 for the full results, in order to use more space to address more questions. We find that ReasonEdit is **very robust to noise injection**. When reasoning facts are replaced with noise, R-Gen and CoE-Gen degrade as expected, because the codebook is missing relevant reasoning facts. We have also added the discussion to a limitation section as follows.
>
> - **W3. Add potential failure cases and limitations.**
>
> Thank you for catching an oversight! We accidentally omitted a limitation paragraph in the submission. We have now added **a detailed Limitations section** to the paper as follows: “(1) ReasonEdit relies on retrieval from a finite codebook, and its performance depends on the coverage of stored keys from observed edits. When a new query image-question pair is far from all stored keys, or when no relevant reasoning facts exist in the codebook, the method will not correct the model’s behavior. (2) In addition, the search over the codebook at inference time can become more expensive as the codebook grows larger, which is a shared limitation by all retrieval-based editors (i.e., GRACE, BalancEdit, IKE). This limitation can be mitigated through efficient implementation of GPU usage and data storage. (3) ReasonEdit assumes users provide accurate and structured reasoning. However, human reasoning can be noisy, incomplete, or incorrect. The current paper provides an initial evaluation of robustness under noisy reasoning, but a more comprehensive study remains an important direction for future work.”
>
> - **Q3. Mechanisms to control large codebook size.**
>
> Currently, ReasonEdit incorporates a key merging mechanism to control codebook size. Specifically, entries with highly overlapping embedding regions are merged, which can reduce storage by 20-25% without hurting performance. Details are provided in Sec D.5 (Table 2). We agree that pruning or forgetting outdated edits is a valuable future direction. Possible extensions to our current mechanism include: (1) usage pruning: remove entries that are rarely retrieved over time, (2) confidence-based filtering: discard entries with low retrieval contribution or inconsistent outcomes, (3) time-decay weighting: gradually downweight older edits to prioritize recent knowledge. We have added this discussion to the paper as a promising future extension. Thank you!
>
> - **Typos.**
>
> Thank you for spotting these, we have fixed the typos. In the revised paper, we added more details of sequential editing in Section 4.4, and we bolded the panel references in the text and enlarged the subtitle font sizes for clarity in Fig 5.

---

### Decision · Program_Chairs · 2026-04-30

**Decision:**

Accept (regular)

**Comment:**

This paper addresses model editing for reasoning-intensive vision–language tasks, which is actually a setting largely overlooked by existing methods. Afterward, it proposes a framework that incorporates human-provided reasoning during editing by storing it in a codebook and retrieving relevant information via a topology-balanced multimodal embedding. Experiments across multiple benchmarks demonstrate the effectiveness of the proposed framework.

It received review comments from four reviewers. After the rebuttal, the three reviewers agreed that they preferred to accept it. One reviewer expressed dissatisfaction with the authors’ feedback. However, after carefully examining both the concerns and the responses, the AC finds that the authors have, in fact, addressed the majority of the issues in a clear and technically adequate manner. While the reviewer’s concerns are understandable, they appear to have been largely resolved in the rebuttal. Accordingly, this reviewer’s assessment is somewhat downweighted, and the work is recommended for acceptance. The comments and suggestions of reviewers should be included and reflected in the final version of this work.